# A First-order Generative Bilevel Optimization Framework for Diffusion Models

**Quan Xiao**[† 1 2]  **Hui Yuan**[3]  **A F M Saif**[1]  **Gaowen Liu**[4]  **Ramana Kompella**[4]  **Mengdi Wang**[3]  **Tianyi Chen**[† 1 2]

## Abstract

Diffusion models, which iteratively denoise data samples to synthesize high-quality outputs, have achieved empirical success across domains. However, optimizing these models for downstream tasks often involves nested bilevel structures, such as tuning hyperparameters for fine-tuning tasks or noise schedules in training dynamics, where traditional bilevel methods fail due to the infinite-dimensional probability space and prohibitive sampling costs. We formalize this challenge as a generative bilevel optimization problem and address two key scenarios: (1) fine-tuning pre-trained models via an inference-only lower-level solver paired with a sample-efficient gradient estimator for the upper level, and (2) training diffusion model from scratch with noise schedule optimization by reparameterizing the lower-level problem and designing a computationally tractable gradient estimator. Our first-order bilevel framework overcomes the incompatibility of conventional bilevel methods with diffusion processes, offering theoretical grounding and computational practicality. Experiments demonstrate that our method outperforms existing fine-tuning and hyperparameter search baselines. Our code has been released at https://github.com/afmsaif/bilevel_diffusion.

## 1. Introduction

Bilevel optimization, which involves nested problems where a *lower-level* optimization is constrained by the solution of

---

[†]This work was done when the authors were at Rensselear Polytechnic Institute. [1]Department of Electrical, Computer, and Systems Engineering, Rensselaer Polytechnic Institute, Troy, NY [2]Department of Electrical and Computer Engineering, Cornell Tech, Cornell University, New York, NY [3]Department of Electrical and Computer Engineering, Princeton University, NJ [4]Cisco Research. Correspondence to: Quan Xiao <quanx1808@gmail.com>, Tianyi Chen <chentianyi19@gmail.com>.

*Proceedings of the 42ⁿᵈ International Conference on Machine Learning*, Vancouver, Canada. PMLR 267, 2025. Copyright 2025 by the author(s).

an *upper-level* objective, has evolved from its theoretical origins in the 1970s (Bracken & McGill, 1973) into a cornerstone of modern machine learning. Its ability to model hierarchical dependencies makes it ideal for complex learning tasks, such as hyperparameter tuning (Pedregosa, 2016), meta-learning (Finn et al., 2017), reinforcement learning (Stadie et al., 2020; Shen et al., 2024), adversarial training (Zhang et al., 2022), neural architecture search (Liu et al., 2019) and LLM alignment (Zakarias et al., 2024; Shen et al., 2025a; Gong et al., 2022; Qin et al., 2024).

Meanwhile, the diffusion model has achieved remarkable success across various domains, particularly in image (Croitoru et al., 2023; Ho et al., 2020; Song et al., 2021a;b), audio (Yang et al., 2023; Liu et al., 2023a), and biological sequence generation (Jing et al., 2022; Wu et al., 2022). Yet, optimizing these models for downstream tasks often introduces nested objectives: for instance, fine-tuning pre-trained models to maximize task-specific rewards (e.g., aesthetic quality (Yang et al., 2024)) risks *reward over-optimization*, where generated samples become unrealistic despite high scores. To mitigate this, incorporating an auxiliary objective can balance the optimization process and prevent excessive focus on a single metric; see Section 3.1. Similarly, designing noise schedules for the forward/reverse processes (Chen, 2023) requires balancing sample quality against computational cost - a problem inherently requiring coordination between training dynamics (lower-level) and scheduler parameters (upper-level); see Section 3.2. An overview of the bilevel generative problem is shown in Figure 1. All of the above illustrate the need for developing bilevel algorithms that are friendly to diffusion models. However, applying existing bilevel methods to *diffusion models* - which operate in infinite-dimensional probability spaces and require costly sampling steps—poses unique challenges. Traditional gradient-based bilevel approaches (Franceschi et al., 2017; Maclaurin et al., 2015) rely on exact lower-level solutions and dense gradient backpropagation, both infeasible for diffusion processes where generating a single sample can involve hundreds of neural network evaluations.

In this paper, we focus on designing computationally efficient and diffusion-friendly approaches for the following problem that we call the *generative bilevel problem*:

$$\min_{x \in \mathcal{X}, y \in \mathcal{P}} f(x, y), \quad \text{s.t.} \quad y \in \mathcal{S}(x) = \arg\min_{y \in \mathcal{P}} g(x, y) \quad (1)$$

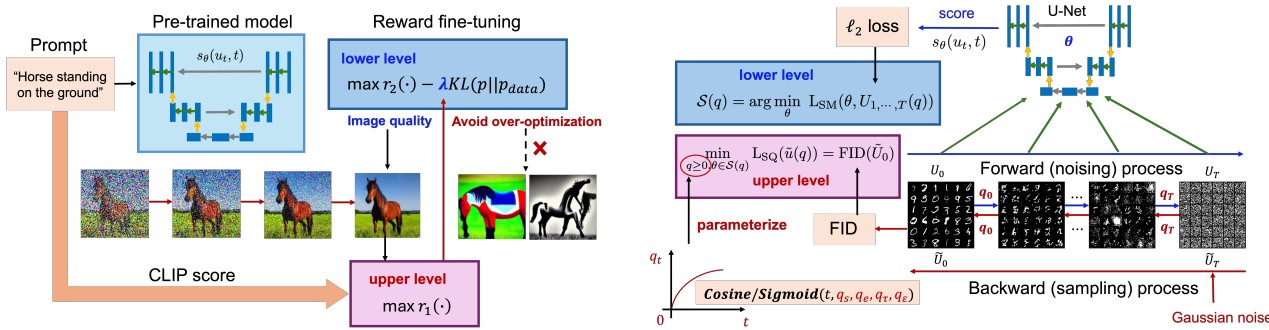

Figure 1. An overview of bilevel generative optimization problems. (Left) Fine-tuning diffusion model with entropy regularization strength parameter $\lambda$. (Right) Noise parameter $q_t$ scheduling problem in the diffusion model.

where $x$ is some hyperparameter in the diffusion model and $y$ represents a distribution we aim to learn, which can be either an image distribution or a noise distribution. Both the upper-level $f : \mathbb{R}^{d_x} \times \mathcal{P} \to \mathbb{R}$ and lower-level objective functions $g : \mathbb{R}^{d_x} \times \mathcal{P} \to \mathbb{R}$ are continuously differentiable, $\mathcal{X} \subset \mathbb{R}^d$ is a closed set, and $\mathcal{P}$ is the probability space. We study the optimistic setting where we select the best response distribution $y \in \mathcal{S}(x)$ to minimize the upper-level loss. Let us denote the minimal lower-level objective value as the value function $g^*(x) = \min_{y \in \mathcal{S}(x)} g(x, y)$ and the nested objective as $F(x) = \min_{y \in \mathcal{S}(x)} f(x, y)$.

The key challenges of solving the *generative bilevel problem* (1) are threefold. First, the lower-level variable $y$ is typically a distribution that operates in infinite dimensional probability space. However, direct access to or optimization over distribution is not feasible; we only have access to samples, and estimating distributions from samples is highly sample-inefficient. Therefore, the gradient over the distribution is inaccessible, making traditional gradient-based bilevel approaches (Shen et al., 2025b; Kwon et al., 2023; Ji et al., 2021; Chen et al., 2021; Hong et al., 2023; Ghadimi & Wang, 2018) generally inapplicable. Second, the objectives $f(x, y)$ and $g(x, y)$ are usually some measures of the sample quality and might not have an explicit form in terms of the hyperparameter $x$, so that $\nabla_x f(x, y)$ and $\nabla_x g(x, y)$ are either not explicitly given or requires sample efficient approximation. Third, for fine-tuning the diffusion model, existing literature related to bilevel fine-tuning on diffusion model (Clark et al., 2024; Marion et al., 2024) requires back-propagations over the pre-trained model, which often suffers from high computational and memory costs. In contrast, we design an *inference-only* bilevel method for this task.

To address these challenges, we classify generative bilevel problems into two categories: (1) those with a pre-trained model, where the target lower-level distribution is the image distribution (e.g., fine-tuning diffusion models), and (2) those without a pre-trained model, where the target lower-level distribution is the noise distribution in the forward process (e.g., noise scheduling during diffusion model train-

ing); see an overview in Figure 1. Two applications we considered are essentially bilevel hyperparameter optimization. To the best of our knowledge, this is the first study to explore bilevel hyperparameter optimization in the context of diffusion models. We develop a *first-order* bilevel framework in both categories to solve (1). The primary differences with non-generative bilevel methods are:

**D1)** for fine-tuning diffusion models that include a pre-trained model, we adopt guidance-based approaches rather than *gradient-based* methods for the lower-level and penalty problems concerning distribution $y$, ensuring the process is training-free and inference-only;

**D2)** for the noise scheduling problem without a pre-trained model, we optimize a noise proxy parameterized by a score neural network, rather than performing noise distribution matching directly in the lower-level problem of (1); and,

**D3)** we design scalable methods to estimate $\nabla_x f(x, y)$ and $\nabla_x g(x, y)$ for two specific applications, i.e. leveraging the closed form of them and proposing sample-efficient estimation for the fine-tuned diffusion model, and using zeroth-order method to estimate $\nabla_x f(x, y)$ in noise scheduling problem without a pre-trained diffusion model.

## 2. Bilevel Optimization and Diffusion Models

In this section, we will review some preliminaries on bilevel optimization, diffusion models, and guided generation, as well as two motivating applications of studying diffusion models in bilevel optimization.

### 2.1. Bilevel optimization

An efficient approach to solving bilevel optimization in (1) is to reformulate (1) to its single-level penalty problem (Shen et al., 2025b; Kwon et al., 2024), given by

$$\min_{x \in \mathcal{X}, y \in \mathcal{P}} \mathcal{L}_\gamma(x, y) := f(x, y) + \gamma(g(x, y) - g^*(x)) \quad (2)$$

and then optimize the upper-level and lower-level variables jointly. By setting the penalty constant $\gamma = \mathcal{O}(\epsilon^{-0.5})$ in-

versely proportional to the target accuracy $\epsilon$, the single-level problem (2) is an $\mathcal{O}(\epsilon)$ approximate problem to the original bilevel problem (1). This method builds upon equilibrium backpropagation (Scellier & Bengio, 2017; Zucchet & Sacramento, 2022), which introduced a similar framework for strongly convex lower-level problems. However, recent works extend the study to accommodate some nonconvex lower-level problems and lay the foundation for broader applications (Shen et al., 2025b; Kwon et al., 2024).

**Gradient-based bilevel approaches.** In the context of diffusion models, the upper-level and lower-level variables, $x$ and $y$, usually have different meanings. To facilitate algorithm design tailored to diffusion models, we decompose (2) into separate $y$- and $x$-optimization problems, that is

$$\min_{x \in \mathcal{X}} \mathcal{L}_\gamma^*(x), \quad \text{with} \quad \mathcal{L}_\gamma^*(x) = \min_{y \in \mathcal{S}_\gamma(x)} \mathcal{L}_\gamma(x, y)$$

$$\text{and} \quad \mathcal{S}_\gamma(x) := \arg\min_{y \in \mathcal{P}} \mathcal{L}_\gamma(x, y). \quad (3)$$

Under some mild conditions specified in (Kwon et al., 2024), $\mathcal{L}_\gamma^*(x)$ is differentiable with the gradient given by

$$\nabla \mathcal{L}_\gamma^*(x) = \nabla_x f(x, z^*) + \gamma(\nabla_x g(x, z^*) - \nabla_x g(x, y^*)) \quad (4)$$

where $z^* \in \mathcal{S}_\gamma^*(x)$ in (3) and $y^* \in \mathcal{S}(x)$ in (1) are any solutions. Moreover, $\nabla \mathcal{L}_\gamma^*(x)$ is a proxy of the original gradient, with the error bounded by $\|\nabla \mathcal{L}_\gamma^*(x) - \nabla F(x)\| \leq \mathcal{O}(1/\gamma)$; see details in Section B. Therefore, when the lower-level problem $g(x, y)$ and the penalty problem $\mathcal{L}_\gamma(x, y)$ are solvable with solutions $y^*, z^*$, we can approximate $\nabla F(x)$ and update the upper-level variable $x$.

### 2.2. Diffusion models

The goal of diffusion models (Song et al., 2021b; Ho et al., 2020; Song et al., 2021a) is to generate samples that match the some target distribution. This is achieved through two complementary stochastic differential equations (SDEs): a forward process, which gradually transforms input data into random noise, and a backward process, which reconstructs the data by denoising the noise. Central to this framework is the learning of a *score function*, which captures the gradient of the log-likelihood of the data distribution and is invariant to the input. This score function, learned during the forward process, is then used to guide the reverse sampling process. Below, we provide a brief overview of these processes.

**Forward process.** The forward process of a diffusion model gradually transforms the original data $U_0 \in \mathbb{R}^D$ into pure noise by incrementally adding noise $dW_t$. This transformation simplifies complex data distributions into a tractable form, enabling efficient modeling and sampling. The following SDE formally describes the process:

$$dU_t = -\frac{1}{2} q(t) U_t \, dt + \sqrt{q(t)} dW_t, \quad \text{for } q(t) > 0 \quad (5)$$

where the initial $U_0$ is a random variable drawn from the data distribution $p_{\text{data}}$, $\{W_t\}_{t \geq 0}$ denotes the standard Wiener process, $q(t)$ is a nondecreasing noise scheduling function, and $U_t$ represents the noise-corrupted data distribution at time $t$. Under mild conditions and for a sufficiently large timestep $T$, (5) transforms the original distribution $p_{\text{data}}$ into a distribution close to Gaussian random noise $\mathcal{N}(0, \mathbf{I}_D)$.

**Backward process.** Given the forward process in (5), the reverse-time SDE is defined by

$$d\widetilde{U}_t = \left[-\frac{1}{2} q(t) \widetilde{U}_t - q(t) \nabla \log p_t \left(\widetilde{U}_t\right)\right] dt$$
$$+ \sqrt{q(t)} d\widetilde{W}_t, \quad \text{for } t \in (0, T], \quad (6)$$

where $d\widetilde{W}_t$ is the reverse-time Wiener process, $p_t(\cdot)$ is the marginal distribution of $U_t$ in the forward process. Let $u_t$ denote the realization of a random variable $U_t$ and $s(u, t) := \nabla \log p_t(u)$ is the score function that has to be estimated in practice or given by the pre-trained model.

**Score matching loss.** To estimate the score function $s(u, t)$, we train a parameterized score network $s_\theta(u, t)$ by tracking the gradient of the log-likelihood of probability from the forward process, which eliminates the need for distribution normalization. Specifically, we minimize the following score-matching loss (Song et al., 2021b):

$$\min_\theta \text{L}_{\text{SM}}(\theta, u)$$
$$:= \mathbb{E}_{t, u_0 \sim p_{\text{data}}, u_t | u_0} \left[\|\nabla \log p_t(u_t | u_0) - s_\theta(u_t, t)\|^2\right] \quad (7)$$

where $t$ is uniformly sampled over the interval $[0, T]$, and $p_t(u_t | u_0)$ is the conditional distribution of $u_t$ over the initial image $u_0$. Typically, the score network $s_\theta(u, t)$ is parametrized by a U-Net model (Ronneberger et al., 2015).

**Guided generation for optimization.** By leveraging the estimated score function $s_\theta(u, T - t)$ from (7) to replace $\nabla \log p_{T-t}(u)$, samples can be generated through the backward process (6). Furthermore, we can introduce guidance terms into the backward process to steer the generation toward the desired reward $V = v$. Using the conditional score function, the goal is to estimate the conditional distribution $\mathbb{P}(U | V = v)$. By Bayes' rule, we have $\nabla_{u_t} \log p_t(u_t \mid v) = \nabla \log p_t(u_t) + \nabla_{u_t} \log p_t(v \mid u_t)$. Given the *pre-trained* score $s_\theta(u, t) \approx \nabla \log p_t(u_t)$ for the unconditional forward process, and the guidance term $G \approx \nabla_{u_t} \log p_t(v \mid u_t)$, the backward SDE is defined by

$$d\widetilde{U}_t = \left[-\frac{1}{2} q(t) \widetilde{U}_t - q(t) s_\theta(\widetilde{U}_t, t) + G(\widetilde{U}_t, t)\right] dt$$
$$+ \sqrt{q(t)} d\widetilde{W}_t, \quad t \in (0, T]. \quad (8)$$

With a proper guidance term and an increasing reward value $v$, the guided sampling process in (8) generates samples that

maximize the given reward function $r(\cdot)$ with an entropy regularization to the pre-trained distribution (Guo et al., 2024; Uehara et al., 2024). The design of the guidance term and the complete procedure of guided generation are outlined in Algorithm 5 and can be found in Appendix E.

## 3. Applications of Generative Bilevel Problems

In this section, we introduce two bilevel optimization problems in diffusion models: one in the fine-tuning stage with a pre-trained model, and another in the pre-training stage.

### 3.1. Reward fine-tuning in diffusion models

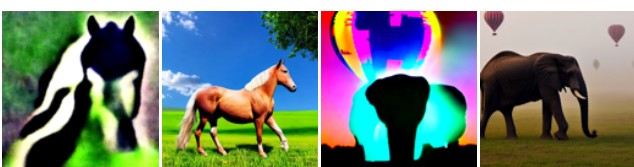

(a) $\lambda = 0.01$   (b) $\lambda = 55.5$   (c) $\lambda = 0.01$   (d) $\lambda = 44.3$

*Figure 2.* Visualization of generated images: (a) Horse and (c) elephant generated with $\lambda = 0.01$, leading to reward over-optimization and resulting in more abstract images misaligned with captions. In contrast, (b) horse and (d) elephant, generated using the bilevel method with $\lambda$ optimized via CLIP score. This suggests CLIP score is a proper metrics for $\lambda$ selection. More visualizations are shown in Figure 8 and can be found in Appendix.

In practice, fine-tuning a pre-trained diffusion model is often necessary to generate samples that achieve high reward. However, if the model over-focuses on reward maximization, it may generate overly aggressive samples that diverge from realistic distributions (Gao et al., 2023). Therefore, a well-tuned model must carefully balance reward optimization with adherence to the pre-trained data distribution. This balance can be formulated as a bilevel optimization problem:

$$\min_{\lambda \in \mathbb{R}_+, p \in \mathcal{S}(\lambda)} f(\lambda, p) := -\mathbb{E}_{u \sim p}[r_1(u)] \qquad (9)$$

$$\text{s.t.} \quad \mathcal{S}(\lambda) = \arg\min_{p' \in \mathcal{P}} \underbrace{-\mathbb{E}_{u \sim p'}[r_2(u)] + \lambda \text{KL}(p' \| p_{\text{data}})}_{g(\lambda, p):=}$$

where the lower level adjusts data distribution by optimizing an entropy regularized reward learning problem (Uehara et al., 2024; Fan et al., 2024), and the upper level selects the best-response entropy strength $\lambda$ by another realistic-measured reward. As shown in Figure 2, the upper-level reward $r_1(\cdot)$ can be caption alignment score (CLIP), further refining the generated samples to align with the provided captions. Since the pre-trained diffusion model is available, $p_{\text{data}}$ is measured by the pre-trained score $s_\theta(u, t)$.

### 3.2. Noise scheduling in diffusion models

Tuning the noise magnitude $q(t)$ in the forward and backward processes is essential for generating high-quality im-

ages with diffusion models. Instead of relying on cross-validation, bilevel optimization can automatically learn an effective noise scheduler, enabling the model to learn useful features while efficiently transforming inputs into noise. In this application, bilevel problem (10) optimizes the noise scheduler in the upper level to minimize a quality score, while the lower level learns the noise distribution from the forward process to match the true Gaussian noise. Instead of framing the lower-level problem as a distribution matching task, we optimize the distribution's parameter $\theta$ as follows.

$$\min_{q \geq 0, \theta \in \mathcal{S}(q)} f(q, \theta) := \mathbb{E}_{\tilde{u}(q) \sim p_\theta}[\text{L}_{\text{SQ}}(\tilde{u}(q))],$$

$$\text{s.t.} \quad \mathcal{S}(q) = \arg\min_{\theta' \in \mathbb{R}^d} g(q, \theta) := \text{L}_{\text{SM}}(\theta', u(q)) \qquad (10)$$

where $p_\theta$ is the probability distribution generated by the backward SDE (6) associated with parameter $\theta$ in the score network, $u(q)$ collects samples in the forward pass from $[0, T]$ defined by schedule $q$, $\text{L}_{\text{SM}}(\cdot)$ is the score matching loss defined in (7), and $\text{L}_{\text{SQ}}(\cdot)$ measures the scheduling quality. For a given schedule $q = \{q(t)\}_{t=1}^T$, at the lower-level, we generate samples $\{u_t\}_{t=1}^T$ according to $q$ and optimize the score network $\theta$ to predict a good proxy of $\nabla \log p_t(u_t)$. Then, at the upper level, we automatically tune the scheduling parameter $q$ by sampling from the reverse process with probability parameterized by $\theta$ and evaluating the quality of the generated samples by $\text{L}_{\text{SQ}}(\cdot)$. Typical choice of $\text{L}_{\text{SQ}}(\cdot)$ can be Fréchet Inception Distance (FID) score, a commonly used metric to measure the quality of the generated image, and is differentiable; see details in Appendix E.3.

## 4. Diffusion-friendly Bilevel Methods

In this section, we first present the meta-algorithm for the generative bilevel problem (1), followed by tailored subroutines for two specific applications in Section 3.

*A meta bilevel algorithm.* To solve the generative bilevel problem (1), we can compute the gradient $\nabla \mathcal{L}_\gamma^*(x)$ according to (4), where the solutions $z^* \in \mathcal{S}_\gamma^*(x)$ and $y^* \in \mathcal{S}(x)$ can be approximated using numerical oracles (e.g., gradient descent or Adam (Kingma, 2015)). Specifically, at each iteration $k$, we first solve the $y$-problem over $\mathcal{L}_\gamma(x^k, y)$ and $g(x^k, y)$ to obtain near-optimal solutions $y^k \approx y^*$ and $z^k \approx z^*$. We then perform gradient descent updates for the $x$-problem according to (4) with $y^k$ and $z^k$. The procedure is detailed in Algorithm 1.

### 4.1. Strategy with pre-trained diffusion models

We first focus on algorithms to guide the generated data distribution of the pre-trained diffusion model toward the optimal solution of (9) in the application of reward fine-tuning diffusion model, with a pre-trained score network.

**Guided-sampling from the lower-level variable.** For

a given $\lambda$, we want to solve two optimization problems $\min_p g(\lambda, p)$ and $\min_p \mathcal{L}_\gamma(\lambda, p)$. For single-level optimization, the distribution generated by the guided backward sampling in Algorithm 5 converges to the optimal distribution for maximizing the reward function $r$ with an entropy regularization term to ensure the generated samples remain close to the pre-training data (Guo et al., 2024). This suggests Algorithm 5 can be used to solve the lower-level problem and penalty problem with respect to $p$. The guidance terms added in the backward sampling process for the lower-level and penalty problems, are defined as

$$G_{\text{lower}}(t, u, \lambda) = \mathrm{G}(u_t, t; r_2)/\lambda \tag{11a}$$

$$G_{\text{penalty}}(t, u, \lambda) = \mathrm{G}(u_t, t; r_1/\gamma + r_2)/\lambda \tag{11b}$$

where $\mathrm{G}(u_t, t; r)$ is defined in (32). Then Algorithm 5 is able to generate samples that are approximately from the optimal lower-level and penalty distribution.

**Gradient update for the upper-level $\lambda$.** By substituting the definition of the objective function into (4), we obtain

$$\nabla \mathcal{L}_\gamma^*(\lambda) = \gamma(\mathrm{KL}(p_\gamma^*(\lambda)\|p_{\text{data}}) - \mathrm{KL}(p^*(\lambda)\|p_{\text{data}})) \tag{12}$$

where $p_\gamma^*(\lambda) \in \arg\min_p \mathcal{L}_\gamma(\lambda, p)$ and $p^*(\lambda) \in \arg\min_p g(\lambda, p)$. Following the gradient-based bilevel method (Kwon et al., 2023; Shen et al., 2025b), a direct way to estimate $\nabla \mathcal{L}_\gamma^*(\lambda)$ is to use guided backward sampling in Algorithm 5 with (11a) and (11b) to obtain samples from $p^*(\lambda)$ and $p_\gamma^*(\lambda)$, and then compute the KL divergence from samples using kernel-based probability estimation. However, it has two drawbacks: 1) each $\lambda$ update requires guided backward sampling, which is computationally expensive; and 2) when the number of samples is less than the dimensionality of the data, the covariance matrix becomes rank-deficient, which makes kernel-based density estimation impossible.

To address these issues, by leveraging the marginal density induced by the SDE, we can derive a closed-form expression for the upper-level gradient in terms of samples.

**Proposition 1.** *The gradient in* (12) *takes the form*

$$\nabla \mathcal{L}_\gamma^*(\lambda) = -\mathbb{E}_{u \sim p_{data}}\left[\lambda^{-1} r_1(u)\right] - \gamma \log \mathbb{E}_{u \sim p_{data}}\left[e^{\frac{r_2(u)}{\lambda}}\right]$$
$$+ \gamma \log \mathbb{E}_{u \sim p_{data}}\left[e^{\frac{r_1(u)/\gamma + r_2(u)}{\lambda}}\right]. \tag{13}$$

**Remark 1.** *This proposition enables us to estimate $\nabla \mathcal{L}_\gamma^*(\lambda)$ directly from the pre-trained distribution, eliminating the need for guided backward sampling in Algorithm 5 to compute $p^*(\lambda)$ and $p_\gamma^*(\lambda)$ for each $\lambda$ and thus, significantly reducing computational cost. The proof is in Appendix D.1.*

Based on Proposition 1, the upper-level gradient in (13) can be estimated via the Monte Carlo method. In other words,

---

**Algorithm 1** A meta generative bilevel algorithm

1: **Inputs:** Initialization $x_0, y_0, z_0$; target error $\epsilon_k$; stepsizes $\eta_k$; penalty constant $\gamma_k$
2: **for** $k = 0, 1, \ldots, K-1$ **do**
3:      estimate $y_k = \arg\min_{y \in \mathcal{P}} g(x^k, y)$ with $\epsilon_k$ error.
4:      estimate $z_k = \arg\min_{z \in \mathcal{P}} \mathcal{L}_{\gamma_k}(x^k, z)$ with $\epsilon_k$ error
5:      estimate $\bar{\nabla}\mathcal{L}_\gamma^*(x_k)$ with $y^k \approx y^*$ and $z^k \approx z^*$ in (4)
6:      update $x_{k+1} = \mathrm{Proj}_{\mathcal{X}}(x_k - \eta_k \bar{\nabla}\mathcal{L}_\gamma^*(x_k))$
7: **end for**
8: **outputs:** $(x_K, z_K)$

---

**Algorithm 2** Bilevel approach with pre-trained model

1: **Input:** Pre-trained score network $s_\theta(\cdot, \cdot)$, differentiable reward $r_1(\cdot), r_2(\cdot)$, stepsizes $\eta_k$, penalty constant $\gamma_k$.
2: sample $\{\tilde{u}_m\}_{m=1}^{M_0}$ from reverse SDE (6) using $s_\theta$
3: **for** $k = 0, 1, \ldots, K-1$ **do**
4:      estimate $\bar{\nabla}\mathcal{L}_{\gamma_k}^*(\lambda_k)$ by (33)
5:      update $\lambda_{k+1} = \mathrm{Proj}_{\mathbb{R}_+}\left(\lambda_k - \eta_k \bar{\nabla}\mathcal{L}_{\gamma_k}^*(\lambda_k)\right)$
6: **end for**
7: sample $\{u_{K,m}^z\}_{m=1}^M$ from Algorithm 5 using $(s_\theta, \frac{r_1}{\gamma_K} + r_2, G_{\text{penalty}}^k(\cdot, \cdot, \lambda_K))$ in (11b)
8: **Output:** $(\lambda_K, \{u_{K,m}^z\}_{m=1}^M)$.

---

with samples $\{\tilde{u}_m\}_{m=1}^{M_0} \sim p_{\text{data}}$, it is estimated by empirical mean with detailed form deferred to Appendix E.2.

The full procedure is summarized in Algorithm 2.

### 4.2. Strategy without pre-trained diffusion models

We next design a bilevel algorithm to solve (10) in the application of noise scheduling for training diffusion models.

**Lower-level problem solver.** When $q$ is given, the samples from the forward process $u(q)$ are determined. Then optimizing the score matching objective $\mathrm{L}_{\text{SM}}(\cdot)$ on the score network gives the optimal lower-level solution, i.e. optimal weights of the score network $\theta \in \arg\min_\theta \mathrm{L}_{\text{SM}}(\theta, u(q))$.

**Penalty problem solver with respect to $\theta$.** Gradient-based approaches on the penalty function $\mathcal{L}_\gamma(q, \theta)$ are effective in solving the penalty problem over $\theta$. The gradient of $\mathcal{L}_\gamma(q, \theta)$ with respect to $\theta$ takes the form of

$$\nabla_\theta \mathcal{L}_\gamma(q, \theta) = \nabla_\theta \mathbb{E}_{\tilde{u}(q) \sim p_\theta}[\mathrm{L}_{\text{SQ}}(\tilde{u}(q))] + \gamma \nabla_\theta \mathrm{L}_{\text{SM}}(\theta, u(q))$$

where the second term can be directly calculated by differentiating the score-matching loss over $\theta$, and the first term can be estimated by the mean of gradients over a batch of samples $\{\tilde{u}_{\theta,q}^m\}_{m=1}^M$ generated by the backward process (6). Although it is possible to obtain the gradient of $\nabla_\theta \mathrm{L}_{\text{SQ}}(\tilde{u}_{\theta,q}^m)$ using PyTorch's auto-differentiation, it requires differentiating through the backward sampling trajectory. Since backward sampling involves 50–100 steps, even for effi-

cient methods like the Denoising Diffusion Implicit Model (DDIM), auto-differentiation is memory-intensive. Instead, we estimate $\nabla_\theta L_{SQ}(\tilde{u}_{\theta,q}^m)$ by zeroth-order (ZO) approximation (Nesterov & Spokoiny, 2017; Shamir, 2017)

$$\widetilde{\nabla}_\theta L_{SQ}(\tilde{u}_{\theta,q}^m) = \frac{\xi}{2\nu}(L_{SQ}(\tilde{u}_{\theta+\nu\xi,q}^m) - L_{SQ}(\tilde{u}_{\theta-\nu\xi,q}^m)) \quad (14)$$

where $\nu > 0$ is the perturbation amount and $\xi \sim \mathcal{N}(0, I_d)$ is randomly drawn from standard Gaussian distribution. At each round, (14) executes two backward processes to get the query of $L_{SQ}(\cdot)$ with two perturbation $\theta + \nu\xi$ and $\theta - \nu\xi$.

**Gradient for the upper-level scheduler $q$.** According to (4), $\nabla\mathcal{L}_\gamma^*(q)$ takes the following form

$$\nabla\mathcal{L}_\gamma^*(q) = \nabla_q \mathbb{E}_{\tilde{u}(q)\sim p_{\theta_z^*}}[L_{SQ}(\tilde{u}(q))]$$
$$+ \gamma(\nabla_q L_{SM}(\theta_z^*, u(q)) - \nabla_q L_{SM}(\theta_y^*, u(q)))$$

where $\theta_y^* \in \arg\min_\theta g(\theta, q)$ and $\theta_z^* \in \arg\min_\theta \mathcal{L}_\gamma(\theta, q)$ are given by the lower-level and penalty problem solver. For $\nabla_q L_{SM}(\theta, u(q))$, according to the chain rule, we have

$$\nabla_q L_{SM}(\theta, u(q)) = \frac{\partial u(q)}{\partial q} \nabla_u L_{SM}(\theta, u(q)). \quad (15)$$

Since the score matching loss for popular choices of diffusion models has explicit dependency on the generated noisy samples $u_t$ in the forward process and $u_t$ has one-step closed form with respect to the initial sample, $\nabla_q L_{SM}(\theta, u(q))$ has closed form for popular choices of diffusion model; see details in Appendix D.2. For $\nabla_q \mathbb{E}_{\tilde{u}(q)\sim p_\theta}[L_{SQ}(\tilde{u}(q))]$ whose explicit expression is not available, we can use similar ZO approaches in (14) with perturbation on noise scheduler $q$ to get the gradient estimator.

**Parametrization for noise scheduler.** To further reduce the memory cost and ensure the nondecreasing of $q(t)$, we parameterize the noise scheduler by the commonly used cosine and sigmoid function (Nichol & Dhariwal, 2021; Chen, 2023; Kingma et al., 2021) and optimize the parameters within these functions instead of directly optimizing $\{q(t)\}_{t=1}^T$. In both parametrization, we optimize just 4 scalar parameters, significantly reducing the dimensionality of the optimization from $T$ to 4. Specifically, we use

$$l(t) = \cos\left[\frac{(t(q_e - q_s) + q_s)/T + q_\epsilon}{1 + q_\epsilon} \times \frac{\pi}{2}\right]^{2q_\tau} \quad (16)$$

where $q_s, q_e, q_\epsilon, q_\tau$ represent the start, end, offset error, and power effect, respectively. Sigmoid parameterization is defined similarly in (34) and can be found in Appendix F.2. Since $q(t)$ should be nondecreasing, we assign $q(t) = 1 - l(t)/l(t-1)$. After parameterization, ZO perturbation will be added on $q_s, q_e, q_\epsilon, q_\tau$ instead of directly on $q(t)$.

The full procedure is summarized in Algorithm 6 and can be found in Appendix E.

## 5. Theoretical Guarantee

In this section, we quantify the theoretical benefits of the bilevel algorithms in both applications. We make the following assumption, which is standard in bilevel optimization literature (Kwon et al., 2023; Ji et al., 2021; Chen et al., 2021; Franceschi et al., 2018; Hong et al., 2023).

**Assumption 1.** *The objective $g(x, \cdot)$ is $\mu_g$-strongly convex, $f(x, y)$ and $g(x, y)$ are jointly smooth over $(x, y)$ with constant $\ell_{f,1}$ and $\ell_{g,1}$ for all $x \in \mathcal{X}$ and $y \in \mathcal{P}$. Moreover, $f(x, \cdot)$ is $\ell_{f,0}$ Lipschitz continuous and $g(x, y)$ has $\ell_{g,2}$ Lipschitz Hessian jointly with respect to $(x, y)$.*

We will justify the validity of this assumption in two generative bilevel applications in Section 3 after we prove the hyperparameter improvement theorem below.

**Theorem 1.** *Under Assumption 1, given any intial hyperparameter $x_0$, letting the inner loop accuracy $\epsilon_k \leq \frac{B}{\gamma_k^2}$, then there exists stepsize $\eta_k \leq \frac{1}{L_F}$ and penalty constant $\gamma_k$ such that the next updates $x_{k+1}$ generated by Algorithm 1 satisfy*

$$F(x_{k+1}) - F(x_k) \leq -\frac{\eta_k}{4}\|G_{\eta_k,\gamma_k}(x_k)\|^2 + \frac{6B^2\eta_k}{\gamma_k^2}$$

*where $G_{\eta,\gamma}(x) = \frac{\text{Proj}_{\mathcal{X}}(x-\eta\bar{\nabla}\mathcal{L}_\gamma^*(x))-x}{\eta}$ is the projected gradient and $L_F, B = \mathcal{O}(1)$ are defined in Lemma 1.*

This theorem demonstrates that when $G_{\eta_k,\gamma_k}(x_k) \neq 0$, and setting penalty constant $\gamma_k \geq \frac{4\sqrt{3}B}{\|G_{\eta_k,\gamma_k}(x_k)\|}$, the bilevel algorithm will have strict descent over the hyper-function $F(x)$. Otherwise, since $\|\bar{\nabla}\mathcal{L}_\gamma^*(x) - \nabla F(x)\| = \mathcal{O}(1/\gamma_k)$, $G_{\eta_k,\gamma_k}(x_k) = 0$ is approximately the stationary points of $F(x)$. The proof can be found in Appendix D.3.

**Implications on generative bilevel applications.** For fine-tuning diffusion models, the KL divergence with respect to a probability distribution $p$ is strongly convex if $p$ is strongly log-concave (Vempala & Wibisono, 2019). This condition is usually met in diffusion models (Song et al., 2021a; Ho et al., 2020), as they generate Gaussian distribution with positive definite covariance matrices. Therefore, when reward function is concave (Guo et al., 2024), Assumption 1 holds. In this application, $\epsilon_k = 0$ since the upper-level gradient estimator does not depend on the inner loop accuracy; see (13). For noise scheduling problem, score matching loss is a composite quadratic function with respect to the noise network (see (24)), which can be viewed as a strongly convex function over the parameterized probability (functional) space (Petrulionyte et al., 2024), so that Assumption 1 holds. In this application, the upper-level gradient estimation error includes an additional term dependent on the error of the ZO estimator, which can be made sufficiently small by appropriately choosing perturbation amount $\nu$.

Therefore, Theorem 1 applies to both settings, indicating that, regardless of initialization - whether from a random

| Baselines | FID ↓ | CLIP ↑ | Time ↓ |
|---|---|---|---|
| Grid search | 125.77 ±1.3 | 31.72 ±2.1 | 3.34 |
| Random search | 123.70 ±4.3 | 35.70 ±1.1 | 3.37 |
| Bayesian search | 140.35 ±2.4 | 33.29 ±2.9 | 12.7 |
| Weighted sum | 116.52 ±3.9 | 36.50 ±1.1 | 4.31 |
| **Bilevel method** | **102.82** ±3.5 | **39.54** ±2.2 | 3.35 |

*Table 1.* Best FID and CLIP score given by different baselines and our method with penalty constant $\gamma = 10^3$ for fine-tuning diffusion model application using synthetic lower-level reward (Yuan et al., 2024). Running time is measured in hours.

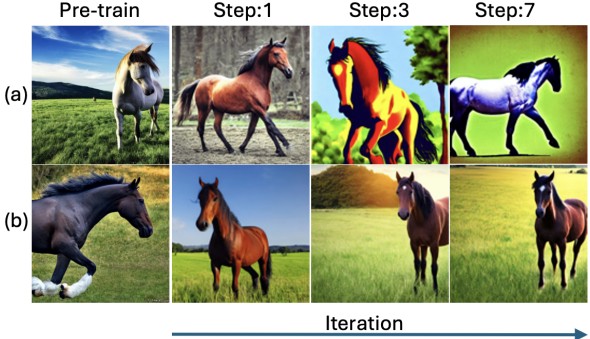

*Figure 3.* Visualization of images generated at different steps: (a) Images generated with $\lambda = 0.1$ become progressively more abstract at each step, while (b) images generated with bilevel method ($\lambda = 55.5$) are more colorful and vivid than the pre-trained images and achieve a perfect balance of quality across steps.

point or selected through cross-validation - the distribution generated using the hyperparameter $x_k$ of bilevel algorithms is guaranteed to perform better. For fine-tuning diffusion models, the bilevel algorithm generates images with higher CLIP scores, while the bilevel noise scheduling algorithm produces images with lower FID scores.

## 6. Numerical Experiments

In this section, we present the experimental results of the proposed bilevel-diffusion algorithms in two applications: reward fine-tuning and noise scheduling for diffusion models, and compare them with baseline hyperparameter optimization methods: grid search, random search, and Bayesian search (Snoek et al., 2012).

### 6.1. Reward fine-tuning in diffusion models

For this experiment, we use the StableDiffusion V1.5 model as our pre-trained model and employ a ResNet-18 architecture (trained on the ImageNet dataset) as the synthetic (lower-level) reward model, following (Yuan et al., 2024), to enhance colorfulness and vibrancy. To enable scalar reward outputs, we replace the final prediction layer of ResNet-18 with a randomly initialized linear layer.

We evaluate bilevel reward fine-tuning Algorithm 5 on the image generation task with complex prompts (Wang et al., 2023; 2024; Clark et al., 2023), comparing it to gradient guidance generation approach in (Guo et al., 2024) combined with conventional hyperparameter search methods for tuning $\lambda$. Inspired by Figure 2, we use CLIP score as the upper-level loss to automatically tune $\lambda$ in a bilevel algorithm. To rule out the impact of the additive effect on the upper-level loss and lower-level reward, we also compare our approach with the weighted sum method, which naively combines the CLIP score and lower-level reward $r_2(\cdot)$, with weight selected by grid search. A detailed description of the baselines is provided in the Appendix F.2.

Table 1 presents the average FID, CLIP score, and execution time for each method over prompts. The bilevel method outperformed traditional hyperparameter tuning methods, achieving superior FID and CLIP scores. Its comparable time complexity arises from requiring backpropagation through the CLIP score, unlike standard methods. For the weighted sum method, which also involves backpropagation through the CLIP score, the bilevel method is faster. Moreover, the bilevel method achieved an $11.76\%$ improvement in the FID score and an $8.32\%$ improvement in the CLIP score over the best-performing weighted sum method.

Figure 3 shows the generated images over generation iteration in Algorithm 5 using $\lambda = 0.1$ and the $\lambda$ optimized by the bilevel approach. The results demonstrate that the entropy strength selected by the bilevel approach achieves a better balance between image quality and realism. Figure 4 shows the optimization process of $\lambda$, revealing that the optimal value of $\lambda$ varies across different prompts.

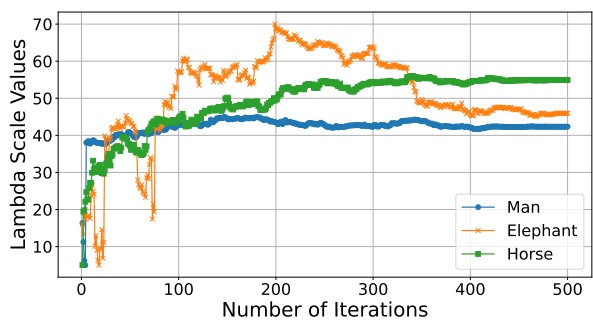

*Figure 4.* Change of $\lambda$ over iteration given by bilevel approach for different prompts.

More visualizations are provided in Figures 9–11 and can be found in Appendix, which visually illustrate the impact of over-aggressive reward optimization, which tends to generate more abstract images (e.g., as observed in the results from grid search and Bayesian search methods).

Furthermore, to showcase the robustness of our approach with respect to different reward functions at the lower level,

| Baselines | FID ↓ | CLIP ↑ | Time ↓ |
|---|---|---|---|
| Grid search | 142.76 ±2.4 | 36.3 ±1.7 | 3.34 |
| Random search | 139.21 ±3.1 | 35.20 ±3.1 | 3.37 |
| Bayesian search | 153.25 ±1.2 | 34.98 ±2.9 | 12.7 |
| Weighted sum | 140.56 ±1.3 | 35.40 ±2.1 | 4.31 |
| **Bilevel method** | **137.23** ±1.7 | **37.10** ±3.2 | 3.35 |

*Table 2.* Best FID and CLIP score given by different baselines and our method with penalty constant $\gamma = 10^3$ for fine-tuning diffusion model application using HPSv2 (Wu et al., 2023) as the lower-level reward. Running time is measured in hours.

we test the performance of each method for benchmarking reward function, HPSv2 (Wu et al., 2023), as the lower-level reward function. Comparisons of our method and different baselines are given in Table 2. Similar to the results given by the synthetic reward function (Yuan et al., 2024), bilevel approach also outperform other baselines in terms of image quality in comparable time complexity. We also provided the visualization of the generated images in Figure 5. While all HPO on fine-tuned models enhance the aesthetic, clarity and sharpness compared to the pre-trained image, the random, grid, Bayesian search, and weighted sum approaches messed up the legs and trunk, and fail to generate the right number of elephant's legs. In comparison, our proposed bilevel approach not only generate colorful images, but also preserve correct elephant biological feature.

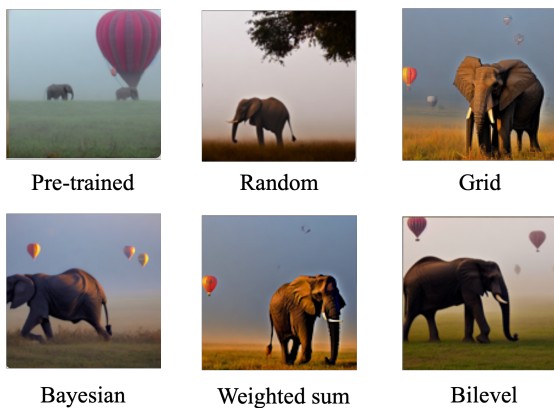

| | | |
|---|---|---|
| Pre-trained | Random | Grid |
| Bayesian | Weighted sum | Bilevel |

*Figure 5.* Visualization of images generated by different methods using the prompt "elephant" and HPSv2 reward (Wu et al., 2023).

### 6.2. Noise scheduling in diffusion models

We evaluated our bilevel noise scheduling method, detailed in Algorithm 6, paired with DDIM backward sampling for the image generation on the MNIST dataset. We trained a U-Net with 178 layers and $10^6+$ parameters following the github repository [1]. We considered both cosine and sigmoid parametrization and tuned 4 parameters $q_s, q_e, q_\tau, q_\epsilon$ jointly.

---

[1] https://github.com/bot66/MNISTDiffusion/tree/main

We compare our method to DDIM combined with baseline hyperparameter optimization methods. We chose greedy grid search over standard grid search as the baseline, as the latter is computationally intensive for searching across multiple hyperparameters. In greedy grid search, we sequentially optimize parameters based on sensitivity, fixing each optimized parameter before tuning the next. Additional experimental setup can be found in Appendix F.2.

Bilevel noise scheduling algorithm in Algorithm 6 alternates between optimizing the weights $\theta$ in U-Net and finding the best noise scheduler $q(t)$ online, so that is computationally efficient and outperforms fixed noise schedulers. Figure 6 shows the learned hyperparameter $q_s, q_e, q_\tau, q_\epsilon$ by bilevel optimization versus iteration $k$ and the corresponding $q(t)$ at four timesteps. We observe the parameters are nontrivial: $q_\tau$ is the most influential factor, varying significantly throughout the training process; start $q_s$ and end $q_e$ update inversely and are utilized more extensively at the beginning of training; and $q_\epsilon$ increases progressively over the course of training. These parameters determine the noise scheduler $q(t)$, which introduces more noise at the beginning and the latter middle stages of training step $k$.

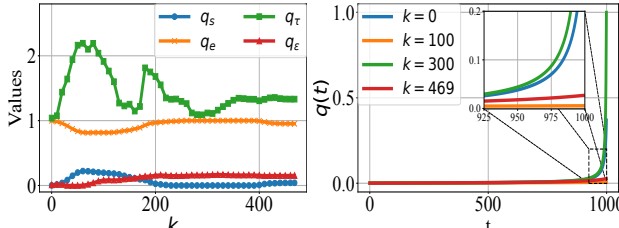

*Figure 6.* Varying hyperparameters start $q_s$, end $q_e$, power $q_\tau$ and offset $q_\epsilon$ in cosine parameterization learned by bilevel method along the training steps and corresponding noise scheduler $q(t)$ at iteration $k = 0, 100, 300, 469$.

Table 3 presents the best FID, inception score (IS), and time complexity achieved by each method. The bilevel method achieves comparable performance with the hyperparameter optimization baselines in both FID and IS while being 6× time faster. Generated images by each method are shown in Figure 7. While the Bayesian approach achieves relatively better FID and IS, it tends to focus on generating simpler numbers, such as 1 and 7. In contrast, the images produced by the bilevel method show excellent diversity across numbers $0 - 9$ while maintaining image fidelity.

## 7. Related works

**Fine-tuning diffusion models.** Fine-tuning diffusion models aims to adapt pre-trained models to boost the reward on downstream tasks. Methods in this domain include directly backpropagating the reward (Clark et al., 2024), RL-based fine-tuning (Fan et al., 2024; Black et al., 2023), direct latent optimization (Tang et al., 2024; Wallace et al., 2023; Hoogeboom et al., 2023), guidance-based approach (Guo

| Methods | Cosine | | | Sigmoid | | |
|---|---|---|---|---|---|---|
| | FID ↓ | IS ↑ | Time ↓ | FID ↓ | IS ↑ | Time ↓ |
| Grid search | 67.30 | 1.76 | 31.53 | 65.31 | 1.76 | 39.12 |
| Random search | 68.97 | 1.61 | 29.62 | 66.02 | 1.69 | 35.94 |
| Bayesian search | 67.16 | 1.69 | 26.85 | 65.17 | 1.65 | 29.13 |
| DDIM (default) | 105.27 | 1.43 | 1.59 | 85.79 | 1.54 | 1.78 |
| **Bilevel method** | **65.41** | **1.78** | 3.88 | **65.16** | **1.79** | 3.94 |

*Table 3.* Comparison of FID, IS, and running time (in hours) for different baselines and our method for the noise scheduling application with cosine and sigmoid parameterization. Default DDIM parameters are from (Nichol & Dhariwal, 2021) for cosine and (Vidhya, 2024) for sigmoid parameterization.

et al., 2024; Chung et al., 2022; Bansal et al., 2023) and optimal control (Uehara et al., 2024). Although entropy regularization is often incorporated into the reward to prevent over-optimization, no existing work has explored designing an efficient bilevel method to tune its strength.

**Noise scheduling in diffusion models.** Noise schedule is crucial to balance the computational efficiency with data fidelity during image generation. Early works, such as DDPM (Ho et al., 2020), employed simple linear schedules for noise variance, while Nichol & Dhariwal (2021) and Kingma et al. (2021) introduced cosine and sigmoid schedules to enhance performance. Recent studies (Lin et al., 2024; Chen, 2023) have highlighted limitations in traditional noise schedules and proposed new parameterization to improve the image quality. Notably, Sahoo et al. (2024) learned the noise scheduler by optimizing the log-likelihood, which yields a tighter lower bound (ELBO) and thus improves the generation quality of the diffusion model. However, none of the prior works considered using bilevel optimization to automatically learn the noise schedule for directly optimizing sample quality.

**Bilevel hyperparameter optimization.** Bilevel optimization has been explored as an efficient hyperparameter optimization framework, including hypernetwork search (Mackay et al., 2019; Liu et al., 2019), hyper-representation (Franceschi et al., 2018), regularization learning (Shaban et al., 2019) and data reweighting (Shaban et al., 2019; Franceschi et al., 2017). Recently, it has been explored in federated learning (Tarzanagh et al., 2022) and LLM fine-tuning (Shen et al., 2025a; Zakarias et al., 2024). None of the existing works have explored hyperparameter optimization in diffusion models, and the methods proposed so far are inapplicable due to the infinite-dimensional probability space and the high computational cost of sampling.

## 8. Conclusions

In this paper, we analyze two types of generative bilevel hyperparameter optimization problems in diffusion models: fine-tuning a diffusion model (with a pre-trained model) and

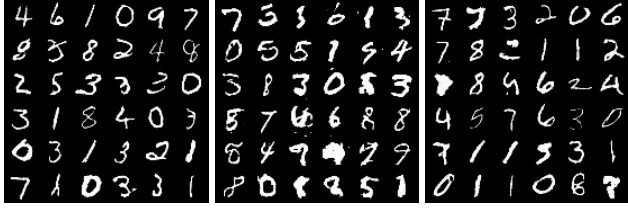

(a) Bilevel          (b) DDIM (default)          (c) Bayesian

*Figure 7.* Visualization of the final generated images by different methods using cosine parameterization.

noise scheduling for training a diffusion model from scratch. For fine-tuning, we propose an inference-only bilevel approach to guide the diffusion model toward the target distribution and leverage the closed-form of KL divergence to update the entropy strength. For training from scratch, we optimize the parameters of the noise distribution to match the true noise and use zeroth-order optimization to determine the optimal noise scheduler for generating high-quality images in the backward process "on the fly." Experiments demonstrate the effectiveness of the proposed method.

## Acknowledgment

The work of Q. Xiao and T. Chen was supported by National Science Foundation (NSF) MoDL-SCALE project 2401297, NSF project 2412486, and the Cisco Research Award.

## Impact Statement

This paper aims to advance diffusion models with bilevel generative optimization, providing a novel approach to hyperparameter tuning for improved image generation. By addressing key challenges in fine-tuning diffusion model and noise scheduling, our work contributes to the broader development of more efficient and adaptive generative models. Potential societal impacts include applications in creative content generation, data augmentation, and machine learning-based simulations. While we acknowledge the possibility of unintended uses, we do not identify any specific societal risks that need to be highlighted in this context.

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

# Appendix for "A First-order Generative Bilevel Optimization Framework for Diffusion Models"

## A. Additional related works

**Bilevel optimization methods.** Bilevel optimization has a long history that dates back to (Bracken & McGill, 1973). Recent efforts have focused on developing efficient gradient-based bilevel optimization methods with non-asymptotic convergence guarantees, inspired by works such as (Ghadimi & Wang, 2018; Ji et al., 2021; Hong et al., 2023; Chen et al., 2021). Since the gradient of the bilevel nested objective depends on the Hessian of the lower-level objective, existing literature proposed different Hessian inversion approximation methods including unrolling differentiation (Franceschi et al., 2017; 2018; Grazzi et al., 2020), implicit differentiation (Chen et al., 2021; Ghadimi & Wang, 2018; Hong et al., 2023; Pedregosa, 2016; Khanduri et al., 2021), conjugate gradients (Ji et al., 2021; Yang et al., 2021) and its warm-started single-loop versions (Arbel & Mairal, 2022; Li et al., 2022; Liu et al., 2023b; Xiao et al., 2023), and equilibrium backpropagation (Scellier & Bengio, 2017; Scellier, 2021); see (Zucchet & Sacramento, 2022) for a comparison. Among these methods, equilibrium backpropagation stands out as a fully first-order approach, valued for its balance of efficiency, robustness, and simplicity. Building on this principle, recent works have extended its applicability from the strongly convex setting to convex, nonconvex and constrained settings by reformulating the bilevel optimization problem as a single-level penalty problem and solving it via first-order approaches (Shen et al., 2025b; Liu et al., 2022; Kwon et al., 2023; 2024; Chen et al., 2024; Lu & Mei, 2023; Jiang et al., 2024; Yao et al., 2024).

## B. Background on bilevel optimization

In this section, we review some background knowledge for first order bilevel optimization.

The differentiability of the penalty problem relies on the differentiability of the value function $g^*(x)$, which is established through the extended Danskin theorem (Shen et al., 2025b, Proposition 4). Specifically, the gradient of value function takes

$$\nabla g^*(x) = \nabla_x g(x, y^*), \ \ \forall y^* \in \mathcal{S}(x). \tag{17}$$

This enables us to solve (2) by gradient-based approach. Similarly, by applying the extended Danskin theorem to the penalty function (Kwon et al., 2024), we know

$$\nabla \mathcal{L}_\gamma^*(x) = \nabla_x \mathcal{L}_\gamma(x, z^*), \quad \text{with} \quad \forall z^* \in \mathcal{S}_\gamma(x)$$

which can be further rewritten according to (17) as

$$\nabla \mathcal{L}_\gamma^*(x) = \nabla_x f(x, z^*) + \gamma(\nabla_x g(x, z^*) - \nabla_x g(x, y^*)). \tag{18}$$

Moreover, the following lemma shows that the penalty objective is a proxy of original bilevel hyper-function $F(x)$.

**Lemma 1** ((Kwon et al., 2023, Lemma 3.1)). *Under Assumption 1, let $\gamma \geq \frac{2\ell_{f,1}}{\mu_g}$, we have $F(x)$ is $L_F$- smooth and*

$$\|\nabla F(x) - \nabla \mathcal{L}_\gamma^*(x)\| \leq \frac{B}{\gamma}$$

*where $L_F = \left(1 + \frac{3l_{g,1}}{\mu_g}\right)\left(l_{f,1} + \frac{l_{g,1}^2}{\mu_g} + \frac{2l_{f,0}l_{g,1}l_{g,2}}{\mu_g^2}\right) = \mathcal{O}(1/\kappa^3)$ and $B = \frac{4l_{f,0}l_{g,1}}{\mu_g^2}\left(l_{f,1} + \frac{2l_{f,0}l_{g,2}}{\mu_g}\right) = \mathcal{O}(1/\kappa^3)$ and $\kappa = \frac{\ell_{f,1}}{\mu_g}$ is the condition number.*

This lemma indicates that $\nabla \mathcal{L}_\gamma^*(x)$ is an approximation of $\nabla F(x)$ with error controlled by enlarging penalty constant $\gamma$.

## C. Background on diffusion models

In this section, we connect continuous to discrete diffusion model to enable the derivation of the closed-form gradient of the score matching function with respect to the noise scheduler in the discrete diffusion model implementation.

Denoising diffusion probabilistic model (DDPM) (Ho et al., 2020) and Denoising diffusion implicit model (DDIM) (Song et al., 2021a) provide standard ways to discretize the continuous SDE in (5) and (6). To be self-contained, we provide a

derivation of connection between them. Let us recall the continuous forward process in (5) as

$$\mathrm{d}U_t = -\frac{1}{2}q(t)U_t\,\mathrm{d}t + \sqrt{q(t)}\mathrm{d}W_t \tag{13}$$

which gives the following transition probabilities

$$p(u_t|u_0) = \mathcal{N}\left(u_0 e^{-\int_0^T \frac{q(s)}{2}\mathrm{d}s}, I\int_0^T q(t)e^{-\int_0^{T-t} q(s)\mathrm{d}s}\mathrm{d}t\right) = \mathcal{N}\left(u_0 e^{-\int_0^T \frac{q(s)}{2}\mathrm{d}s}, \left(1 - e^{-\int_0^T q(s)\mathrm{d}s}\right)I.\right)$$

See also (Song et al., 2021b, Appendix B) and (Denker et al., 2024, Appendix A). Therefore, by defining $\bar{q}(t) = e^{-\int_0^t q(s)\mathrm{d}s}$, we get the form in DDPM (Ho et al., 2020)

$$p(u_t|u_0) = \mathcal{N}\left(\sqrt{\bar{q}(t)}u_0, (1 - \bar{q}(t))\mathbf{I}_d\right). \tag{19}$$

Since the approximation $1 - x \approx e^{-x}$ holds well when $x$ is small, we have a discrete approximation of $\bar{q}(t)$ as

$$\bar{q}(t) = e^{-\int_0^T q(s)ds} \approx \prod_{n=0}^{N-1}\left(1 - q(t_n)\Delta t\right).$$

By choosing $\Delta t = 1$, we get the expression of discrete DDPM in (Ho et al., 2020) as follows.

**Forward process.** Given a data point sampled from a data distribution $u_0 \sim p_{\mathrm{data}}$, the forward process in DDPM generates a sequence of samples $u_1, u_2, \ldots, u_T$ by gradually adding noise

$$p(u_{1:T}|u_0) = \prod_{t=1}^T p(u_t|u_{t-1}), \quad p(u_t|u_{t-1}) = \mathcal{N}(\sqrt{1 - q_t}u_{t-1}, q_t\mathbf{I}_d) \tag{20}$$

where $\{q_t\}_{t=1}^T$ corresponds to the noise scheduler in discrete DDPM. (20) can be further expressed as

$$p(u_t|u_0) = \mathcal{N}\left(\sqrt{\bar{q}_t}u_0, (1 - \bar{q}_t)\mathbf{I}_d\right) \tag{21}$$

where $\bar{q}_t = \prod_{s=1}^t(1 - q_s)$ is the variance scheduler defined by the noise scheduler $\{q_t\}_{t=1}^T$.

**Backward process of DDPM.** The backward process aims to recover $u_0$ from $u_T$ by iteratively denoising

$$\tilde{p}_\theta(u_{0:T}) = \tilde{p}(u_T)\prod_{t=1}^T \tilde{p}_\theta(u_{t-1}|u_t), \tag{22}$$

where $\tilde{p}(u_T) = \mathcal{N}(0, \mathbf{I}_d)$ and each $p_\theta$ is modeled as a Gaussian distribution parameterized by $\theta$

$$\tilde{p}_\theta(u_{t-1}|u_t) = \mathcal{N}\left(\mu_\theta(u_t, t), \sigma_\theta^2(u_t, t)\mathbf{I}_d\right)$$

with the mean and variance learned by optimizing the score-matching objective.

**Score matching.** In the discrete DDPM, score-matching loss also takes a simpler form. To learn $\mu_\theta$ and $\sigma_\theta$, we first estimate the backward probability given the initial state using the Gaussian kernel estimation as follows

$$\tilde{p}_t(u_{t-1}|u_t, u_0) = \mathcal{N}\left(\mu_t(u_t, u_0), \sigma_t^2\mathbf{I}_d\right)$$

$$\text{where} \quad \mu_t(u_t, u_0) = \frac{\sqrt{\bar{q}_{t-1}}q_t}{1 - \bar{q}_t}u_0 + \frac{\sqrt{1 - q_t}(1 - \bar{q}_{t-1})}{1 - \bar{q}_t}u_t \stackrel{(a)}{=} \frac{1}{\sqrt{1 - q_t}}\left(u_t - \frac{q_t}{\sqrt{1 - \bar{q}_t}}\delta_t\right) \tag{23a}$$

$$\text{and} \quad \sigma_t^2 = \frac{1 - \bar{q}_{t-1}}{1 - \bar{q}_t}q_t \tag{23b}$$

where **(a)** is earned by reparameterizing (21) as $u_t(u_0, \delta_t) = \sqrt{\bar{q}_t}u_0 + (1 - \bar{q}_t)\delta_t$ for $\delta_t \sim \mathcal{N}(0, \mathbf{I}_d)$. As $\mu_t$ is proportional to $\delta_t$, we can fit a neural network to proxy $\mu_t$ by optimizing the score matching loss in (7) in the following simplified form with explicit dependence on noise scheduler $q$

$$\mathrm{L}_{\mathrm{SM}}(\theta, q) = \mathbb{E}_{u_0, \delta, t}\left[\left\|\delta - \delta_\theta\left(\sqrt{\bar{q}_t}u_0 + \sqrt{1 - \bar{q}_t}\delta, t\right)\right\|^2\right] \tag{24}$$

where $\delta_\theta$ is a neural network approximator (e.g. U-Net) intended to predict Gaussian noise $\delta$ from $u_t$.

By optimizing the score matching objective $L_{\text{SM}}(\theta, q)$ with respect to $\theta$, we obtain the proxy of $\delta_\theta$ and using $\delta_\theta$ instead of $\delta_t$ in (23a), we can sample the backward process by

$$u_{t-1} = \frac{1}{\sqrt{q_t}}\left(u_t - \frac{1-q_t}{\sqrt{1-\bar{q}_t}}\delta_\theta\right) + \sigma_t v \tag{25}$$

with $v \sim \mathcal{N}(0, \mathbf{I}_d)$. The full training and backward sampling process in DDPM is summarized in Algorithm 3 and 4.

---

**Algorithm 3** Score network training

1: **repeat**
2:     draw $\{u_0^m\}_{m=1}^M \sim p_{\text{data}}$
3:     $\{t_m\}_{m=1}^M \sim \text{Uniform}([T])$
4:     $\{\delta_m\}_{m=1}^M \sim \mathcal{N}(0, \mathbf{I}_d)$
5:     Take gradient descent step on $\nabla_\theta \frac{1}{M}\sum_{m=1}^M \|\delta - \delta_\theta(\sqrt{\bar{q}_{t_m}}u_0^m + \sqrt{1-\bar{q}_{t_m}}\delta_m, t_m)\|^2$
6: **until** converged

---

**Algorithm 4** Backward sampling

1: $\{\tilde{u}_T^m\}_{m=1}^M \sim \mathcal{N}(0, \mathbf{I}_d)$
2: **for** $t = T, \ldots, 1$ **do**
3:     $\{v^m\}_{m=1}^M \sim \mathcal{N}(0, \mathbf{I}_d)$ if $t > 1$, else $v^m = 0$
4:     $\tilde{u}_{t-1}^m = \frac{1}{\sqrt{1-q_t}}\left(\tilde{u}_t^m - \frac{q_t}{\sqrt{1-\bar{q}_t}}\delta_\theta(\tilde{u}_t^m, t)\right) + \sigma_t v^m$
5: **end for**
6: **return** $\frac{1}{M}\sum_{m=1}^M u_0^M$

---

DDIM (Song et al., 2021a) uses the same forward process and score network training as DDPM, but employs a deterministic backward sampling strategy and eliminates redundant sampling steps to further accelerate the backward process as follows.

**Backward process of DDIM.** Letting $\{t_i\}$ be some selected time steps from $[0, T]$, (25) is generalized by

$$u_{t_{i-1}} = \sqrt{\bar{q}_{t_{i-1}}}\left(\frac{u_{t_i} - \sqrt{1-\bar{q}_{t_i}}\,\delta_\theta^{(t_i)}(u_{t_i})}{\sqrt{\bar{q}_{t_i}}}\right) + \sqrt{1-\bar{q}_{t_{i-1}} - \sigma_{t_i}^2}\cdot \delta_\theta^{(t_i)}(u_{t_i}) + \sigma_{t_i}v_{t_i}, \tag{26}$$

which recovers DDPM in (25) when $\sigma_t = \sqrt{(1-\bar{q}_{t-1})q_t/(1-\bar{q}_t)}$ in (23b) and without skipping. i.e. $t_i = t$, and the resulting deterministic model when $\sigma_t = 0$ is called DDIM. In DDIM, time steps $\{t_i\}$ are selected using either linear ($t_i = \lfloor ci \rfloor$ for some $c$) or a quadratic ($t_i = \lfloor ci^2 \rfloor$ for some $c$) strategy. With these designs, backward sampling steps of DDIM can be reduced from 1000 in DDPM to $50 - 10$ (Song et al., 2021a).

# D. Theoretical analysis

In this section, we present the closed-form of $\nabla \mathcal{L}_\gamma^*(\lambda)$ for fine-tuning diffusion model application, the gradient of the score matching function with respect to noise scheduler in discrete diffusion models, and the theoretical guarantee of Algorithm 1.

### D.1. Upper-level gradient: Proof for Proposition 1

**Proof:** According to (Tang, 2024, Equation (3.7)), we have the closed form of KL divergence of fine-tuning distribution and pre-trained distribution as follows.

$$\text{KL}\left(p^*(\lambda)\|p_{\text{data}}\right) = -\mathbb{E}_{u\sim p_{\text{data}}}\left[\frac{r_2(u)}{\lambda}\right] + \log \mathbb{E}_{u\sim p_{\text{data}}}\left[e^{r_2(u)/\lambda}\right]$$

$$\text{KL}\left(p_\gamma^*(\lambda)\|p_{\text{data}}\right) = -\mathbb{E}_{u\sim p_{\text{data}}}\left[\frac{r_1(u)/\gamma + r_2(u)}{\lambda}\right] + \log \mathbb{E}_{u\sim p_{\text{data}}}\left[e^{\frac{r_1(u)/\gamma + r_2(u)}{\lambda}}\right] \tag{27}$$

Then the proof can be obtained by plugging the closed-form of KL divergence in (27) into (12). That is,

$$\nabla \mathcal{L}_\gamma^*(\lambda) = \gamma(\text{KL}(p_\gamma^*(\lambda)\|p_{\text{data}}) - \text{KL}(p^*(\lambda)\|p_{\text{data}}))$$

$$= -\mathbb{E}_{u\sim p_{\text{data}}}\left[\frac{r_1(u)/\gamma + r_2(u)}{\lambda}\right] + \log \mathbb{E}_{u\sim p_{\text{data}}}\left[e^{\frac{r_1(u)/\gamma + r_2(u)}{\lambda}}\right] + \mathbb{E}_{u\sim p_{\text{data}}}\left[\frac{r_2(u)}{\lambda}\right] - \log \mathbb{E}_{u\sim p_{\text{data}}}\left[e^{r_2(u)/\lambda}\right]$$

$$= -\mathbb{E}_{u\sim p_{\text{data}}}\left[\lambda^{-1}r_1(u)\right] - \gamma \log \mathbb{E}_{u\sim p_{\text{data}}}\left[e^{\frac{r_2(u)}{\lambda}}\right] + \gamma \log \mathbb{E}_{u\sim p_{\text{data}}}\left[e^{\frac{r_1(u)/\gamma + r_2(u)}{\lambda}}\right]$$

which completes the proof.

### D.2. Explicit lower-level noise scheduler's gradient in discrete diffusion models

In this section, we derive the explicit gradient expression of the score matching function with respect to the noise scheduler.

Since score matching loss in both DDPM and DDIM takes the form in (15), the noise scheduler's gradient in score matching objective can be earned by the chain rule

$$\nabla_q \mathrm{L}_{\mathrm{SM}}(\theta, u(q)) \approx \frac{1}{M} \sum_{m=1}^{M} \frac{\partial u_q^m}{\partial q} \nabla_u \mathrm{L}_{\mathrm{SM}}(\theta, u_q^m) \tag{28}$$

where $u_q^m = u_{t_m}^m$ with $t_m$ uniformly chosen from $t \in [T] = \{1, \cdots, T\}$ and subscript $q$ means this forward sample is generated using noise scheduler $q = [q_1, \cdots, q_T]$. In this way, using the reparameterization $u_t = \sqrt{\bar{q}_t} u_0 + (1 - \bar{q}_t)\delta$ and the relation of $\bar{q}_t = \prod_{s=1}^{t}(1 - q_s)$, we have for any $t \le t_m$,

$$\frac{\partial u_q^m}{\partial q_t} = \frac{\partial u_{t_m}^m}{\partial q_t} = \frac{\partial \bar{q}_{t_m}}{\partial q_t} \frac{\partial u_{t_m}^m}{\partial \bar{q}_{t_m}} = -\frac{\bar{q}_{t_m}}{q_t} \left( \frac{u_0}{2\sqrt{\bar{q}_{t_m}}} - \delta \right)^{\top}. \tag{29}$$

On the other hand, the gradient of the score-matching function with respect to sample $u$ takes the form of

$$\nabla_u \mathrm{L}_{\mathrm{SM}}(\theta, u_q^m) = \frac{\partial \delta_\theta(u_{t_m}^m)}{\partial u_{t_m}^m} \frac{\partial \mathrm{L}_{\mathrm{SM}}(\theta, u_q^m)}{\partial \delta_\theta(u_{t_m}^m)} = 2 \frac{\partial \delta_\theta(u_{t_m}^m)}{\partial u_{t_m}^m} (\delta_\theta(u_{t_m}^m) - \delta_m) \tag{30}$$

where $u_{t_m}^m = \sqrt{\bar{q}_{t_m}} u_0^m + (1 - \bar{q}_{t_m})\delta_m$ and the first term $\frac{\partial \delta_\theta(u_{t_m}^m)}{\partial u_{t_m}^m}$ is the derivative of the score network with respect to the input samples that is directly obtainable via auto-differentiation library in Pytorch. By plugging the above closed forms of partial derivative in (29) and (30) into (28), we get for any $t \in [T]$,

$$\nabla_{q_t} \mathrm{L}_{\mathrm{SM}}(\theta, u(q)) \approx \frac{1}{|\mathcal{M}_t|} \sum_{\mathcal{M}_t := \{m | t_m \ge t\}} \frac{\partial u_q^m}{\partial q_t} \nabla_u \mathrm{L}_{\mathrm{SM}}(\theta, u_q^m)$$

$$= \frac{2}{|\mathcal{M}_t|} \sum_{\mathcal{M}_\tau := \{m | t_m \ge t\}} -\frac{\bar{q}_{t_m}}{q_t} \left( \frac{u_0}{2\sqrt{\bar{q}_{t_m}}} - \delta_m \right)^{\top} \frac{\partial \delta_\theta(u_{t_m}^m)}{\partial u_{t_m}^m} (\delta_\theta(u_{t_m}^m) - \delta_m). \tag{31}$$

In practice, we do not need to manually implement the closed form in (31), as PyTorch's auto-differentiation handles it automatically. The derivation in this section highlights the low computational cost of auto-differentiation, as only $\frac{\partial \delta_\theta(u_{t_m}^m)}{\partial u_{t_m}^m}$ depends on the U-Net structure, and this differential is commonly used in gradient guidance diffusion models (Guo et al., 2024; Bansal et al., 2023).

### D.3. Descent theorem: Proof of Theorem 1

**Proof:** By Taylor expansion and the $L_F$ smoothness of $F(x)$, we have

$$F(x_{k+1}) \le F(x_k) + \langle \nabla F(x_k), x_{k+1} - x_k \rangle + \frac{L_F}{2} \|x_{k+1} - x_k\|^2$$

$$\le F(x_k) + \langle \nabla F(x_k), \mathrm{Proj}_{\mathcal{X}}(x_k - \eta_k \bar{\nabla} \mathcal{L}_{\gamma_k}^*(x_k)) - x_k \rangle + \frac{L_F}{2} \| \mathrm{Proj}_{\mathcal{X}}(x_k - \eta_k \bar{\nabla} \mathcal{L}_{\gamma_k}^*(x_k)) - x_k \|^2$$

$$= F(x_k) + \langle \bar{\nabla} \mathcal{L}_{\gamma_k}^*(x_k), \mathrm{Proj}_{\mathcal{X}}(x_k - \eta_k \bar{\nabla} \mathcal{L}_{\gamma_k}^*(x_k)) - x_k \rangle + \frac{L_F}{2} \| \mathrm{Proj}_{\mathcal{X}}(x_k - \eta_k \bar{\nabla} \mathcal{L}_{\gamma_k}^*(x_k)) - x_k \|^2$$

$$+ \langle \nabla F(x_k) - \bar{\nabla} \mathcal{L}_{\gamma_k}^*(x_k), \mathrm{Proj}_{\mathcal{X}}(x_k - \eta_k \bar{\nabla} \mathcal{L}_{\gamma_k}^*(x_k)) - x_k \rangle$$

$$\le F(x_k) + \langle \bar{\nabla} \mathcal{L}_{\gamma_k}^*(x_k), \mathrm{Proj}_{\mathcal{X}}(x_k - \eta_k \bar{\nabla} \mathcal{L}_{\gamma_k}^*(x_k)) - x_k \rangle + \frac{L_F}{2} \| \mathrm{Proj}_{\mathcal{X}}(x_k - \eta_k \bar{\nabla} \mathcal{L}_{\gamma_k}^*(x_k)) - x_k \|^2$$

$$+ \frac{1}{2\alpha} \|\nabla F(x_k) - \bar{\nabla} \mathcal{L}_{\gamma_k}^*(x_k)\|^2 + \frac{\alpha}{2} \| \mathrm{Proj}_{\mathcal{X}}(x_k - \eta_k \bar{\nabla} \mathcal{L}_{\gamma_k}^*(x_k)) - x_k \|^2$$

$$\overset{(a)}{\le} F(x_k) - \frac{1}{4\eta_k} \| \mathrm{Proj}_{\mathcal{X}}(x_k - \eta_k \bar{\nabla} \mathcal{L}_{\gamma_k}^*(x_k)) - x_k \|^2 + \eta_k \|\nabla F(x_k) - \bar{\nabla} \mathcal{L}_{\gamma_k}^*(x_k)\|^2$$

$$\overset{(b)}{\leq} F(x_k) - \frac{1}{4\eta_k} \| \operatorname{Proj}_{\mathcal{X}}(x_k - \eta_k \bar{\nabla} \mathcal{L}^*_{\gamma_k}(x_k)) - x_k \|^2 + \frac{2B^2 \eta_k}{\gamma_k^2} + 4\eta_k \gamma_k^2 \epsilon_k^2$$

where $(a)$ comes from the descent lemma of projected gradient (e.g. (Nesterov et al., 2018, Theorem 2.2.13)) and choosing $\alpha = \frac{1}{2\eta_k}$ and $(b)$ is because

$$\|\nabla F(x_k) - \bar{\nabla} \mathcal{L}^*_{\gamma_k}(x_k)\|^2 \leq 2\|\nabla F(x_k) - \nabla \mathcal{L}^*_{\gamma_k}(x_k)\|^2 + 2\|\bar{\nabla} \mathcal{L}^*_{\gamma_k}(x_k) - \nabla \mathcal{L}^*_{\gamma_k}(x_k)\|^2 \leq \frac{2B^2}{\gamma_k^2} + 4\gamma_k^2 \epsilon_k^2$$

where the last inequality comes from Lemma 1 and the estimation error $\epsilon_k$ of penalty and lower-level problem. By defining the projected gradient as $G_{\eta,\gamma}(x) = \frac{\operatorname{Proj}_{\mathcal{X}}(x - \eta \bar{\nabla} \mathcal{L}^*_{\gamma}(x)) - x}{\eta}$ and letting $\epsilon_k \leq \frac{B}{\gamma_k^2}$, we get the conclusion.

# E. Complete Algorithms

In this section, we present the complete algorithms with additional details for gradient guided diffusion model for (single-level) generative optimization (Guo et al., 2024), and the bilevel diffusion algorithm for fine-tuning and noise scheduling problem proposed in this work.

## E.1. Guided Diffusion algorithm for Generative Optimization

To generate samples that optimize a given reward function $r$, we can iteratively apply the backward SDE in (6) with the pre-trained score network $s_\theta$ and the guidance defined as

$$G(\tilde{u}_t, t; r) = -\rho(t) \nabla_{\tilde{u}_t} \left[ v - \frac{g^\top ((\tilde{u}_t + h(t) s_\theta(\tilde{u}_t, t)))}{\sqrt{\bar{q}(t)}} \right]^2 \tag{32}$$

where $g$ is a gradient vector associated with the reward function $r(\cdot)$ evaluated at the current sample $\tilde{u}_t$, $v$ is a given target reward value that increase along the optimization, $\bar{q}(t) = \exp(-\int_0^t q(s)ds)$, $h(t) = 1 - \bar{q}(t)$ are the mean and variance of $t$-th sample and $\rho(t)$ is the tunning parameter.

We can iteratively update the gradient guidance to steer the sample generation process maximize the reward function. Specifically, at each iteration $n$, the backward SDE (8) is stimulated using the current gradient guidance from (32), evaluated at the current samples, to generate new samples. Subsequently, the gradient guidance term is updated at the newly generated samples. After $N$ steps of guidance updates, we are able to generate samples approximately that follow the target distribution with $\mathcal{O}(\log(1/N))$, effectively minimizing $r(\cdot)$ while incorporating regularization to align with the pre-trained model (Guo et al., 2024). The complete algorithm for guided diffusion model for generative optimization is outlined in Algorithm 5.

## E.2. Bilevel fine-tuning algorithm

Bilevel fine-tuning diffusion algorithm is summarized in Algorithm 2 with the following upper-level gradient estimation.

**Monte Carlo estimation of upper-level gradient.** The upper-level gradient can be estimated from the following way

$$\bar{\nabla} \mathcal{L}^*_\gamma(\lambda) = -\frac{1}{\lambda M_0} \sum_{m=1}^{M_0} r_1(\tilde{u}_m) - \gamma \log \frac{1}{M_0} \sum_{m=1}^{M_0} \left[ e^{\frac{r_2(\tilde{u}_m)}{\lambda}} \right] + \gamma \log \frac{1}{M_0} \sum_{m=1}^{M_0} \left[ e^{\frac{r_1(\tilde{u}_m)/\gamma + r_2(\tilde{u}_m)}{\lambda}} \right]. \tag{33}$$

where $\{\tilde{u}_m\}_{m=1}^{M_0}$ are samples from pre-trained distribution.

## E.3. Bilevel noise scheduling algorithm

With ZO estimation and the parametrization, the complete algorithm for bilevel noise scheduling problem is summarized in Algorithm 6. Besides, the upper-level schedule quality loss is differentiable according to (Mathiasen & Hvilshøj, 2020).

**Differentiable $L_{SQ}(\cdot)$ loss.** FID score is a commonly used metric in computer vision to measure the distance of the generated distribution and the true distribution. Given $\{u_m\}_{m=1}^M \sim p_\theta$ generated from diffusion model algorithm and $\{\tilde{u}_m\}_{m=1}^{M_0} \sim p_{\text{data}}$, we first encode all samples $u_m, \tilde{u}_m$ by the pre-trained Inception network (Seitzer, 2020) and then FID

---

**Algorithm 5** Guided Diffusion for Generative Optimization

---

1: **Input:** Pre-trained score network $s_\theta(\cdot, \cdot)$, differentiable reward $r(\cdot)$, guidance $G$.
2: **Parameter:** Strength parameters $\rho(t)$, $\{v_n\}_{n=0}^{N-1}$, number of iterations $N$, batch sizes $\{B_n\}$.
3: **Initialization:** $G_0 = \text{NULL}$.
4: **for** iteration $n = 0, \ldots, N-1$ **do**
5:    **Generate:** Sample $\tilde{u}_{n,i}$ for $i \in [B_n]$ by backward SDE in (8) with $(s_\theta, G_n)$ until time $T$
6:    **Compute Guidance:**

    (i)    Sample mean $\bar{u}_n := \frac{1}{B_n} \sum_{i=1}^{B_n} \tilde{u}_{n,i}$.

    (ii)   Query gradient $g_n = \nabla r(\bar{u}_n)$.

    (iii)  Update gradient guidance $G_{n+1}(\cdot, \cdot) = \text{G}(\cdot, \cdot; r)$ via (32), using $s_\theta$, gradient vector $g_n$, and reward target $v_n$
           and $\beta(t)$.

7: **end for**
8: **Generate:** Sample $\tilde{u}_i$ for $i \in [B_N]$ by backward SDE in (8) with $(s_\theta, G_N)$ until time $T$
9: **Output:** $\{\tilde{u}_i\}_{i=1}^{B_N}$.

---

score is computed by the Wasserstein distance between the two multivariate normal distributions. Therefore, when the Inception network used for encoding is differentiable with respect to its input, as the one proposed by Seitzer (2020) does, $\text{L}_{\text{SQ}}(\cdot)$ is differentiable with respect to the sample and then by the chain rule, it is differentiable with respect to $q$ and $\theta$.

# F. Experimental details

In this section, we introduce the details of experimental setup for two applications. All experiments were conducted on two servers: one with four NVIDIA A6000 GPUs, and 256 GB of RAM; one with an Intel i9-9960X CPU and two NVIDIA A5000 GPUs.

## F.1. Application 1: fine-tuning diffusion model with bilevel entropy regularization learning

For the hyperparameter settings, we set the initial value of noise scheduler $q(t)$ to 1 and tune the entropy strength $\lambda$ using different methods. We use a batch size of 3 for the fine-tuning step and set optimization step 7 and repeat the optimization for 4 times. The prompts we used for generating these figures are mentioned in the figure $9 - 11$. We compare our results against various baseline methods including grid search, random search, bayesian method, and weighted sum.

**Grid Search.** For the grid search method, we selected the following $\lambda$ values and conducted simulations for each value:

$$\lambda \in \{0.01, 0.1, 1.0, 10.0, 100\}$$

**Random search.** We fine-tuned the diffusion model using $\lambda$ values generated by a random value generator. Specifically, we sampled 5 random values logarithm uniformly from the range $[0.01, 100]$. The $\lambda$ values generated are as follows:

$$\lambda \in \{62.3, 74.0, 74.18, 79.52, 94.25\}$$

**Bayesian search.** We utilized Bayesian optimization with the objective of maximizing the reward function and the CLIP score. Specifically, the search space for the hyperparameter $\lambda_{\text{scale}}$ was defined as a continuous range $[0.01, 100]$, sampled on a logarithmic scale using a log-uniform distribution. The optimization process was conducted using the `gp_minimize` function from the `scikit-optimize` library, which employs Gaussian process-based Bayesian optimization. To balance computational efficiency and optimization quality, the number of function evaluations was limited to $n_{\text{calls}} = 15$. Additionally, a fixed random seed (`random_state = 42`) was set to ensure the reproducibility of results.

Since Bayesian optimization minimizes the objective function by default, we reformulated the problem by negating the combined reward and CLIP score, thereby transforming the maximization task into a minimization problem. This reformulation allowed us to identify the optimal $\lambda_{\text{scale}}$ value that best balances reward maximization and adherence to the

---

**Algorithm 6** Bilevel Approach without Pre-trained Diffusion Model

---

1: **Input:** Differentiable loss functions $\mathrm{L}_{\mathrm{SQ}}(\cdot)$ and $\mathrm{L}_{\mathrm{SM}}(\cdot)$, initial samples $\{\tilde{u}_m\}_{m=1}^{M_0}$, iteration number $K, S_z, S_y$, initial noise scheduler parametrization parameter $q_{\mathrm{param}} = \{q_s, q_e, q_\tau, q_\epsilon\}$ (cosine or sigmoid), feasible set for $q_{\mathrm{param}} \in \mathcal{Q}$, stepsizes $\beta, \eta_k$.

2: **for** $k = 0, 1, \ldots, K-1$ **do**

3:     sample $\{u_{k,m}\}_{m=1}^M$ from the forward process (5) with noise scheduler $q_k$.         $\triangleright q_{0,\mathrm{param}} = q_{\mathrm{param}}$

4:     **for** $s = 0, 1, \ldots, S_y^k - 1$ **do**

5:         update $\theta_{k,s+1}^y = \theta_{k,s}^y - \frac{\beta}{M} \sum_{m=1}^M \nabla_\theta \mathrm{L}_{\mathrm{SM}}(\theta_{k,s}^y, u_{k,m})$.         $\triangleright \theta_{k,0}^y = \theta_k^y, \theta_{k+1}^y = \theta_{k,S_y}^y$

6:     **end for**

7:     **for** $s = 0, 1, \ldots, S_z - 1$ **do**

8:         sample $\{\tilde{u}_{k,m}^{s,+}, \tilde{u}_{k,m}^{s,-}\}_{m=1}^M$ from (6) with $q_k$ and $\theta_{k,s}^z + \nu\theta_{\mathrm{perturb}}$ and $\theta_{k,s}^z - \nu\theta_{\mathrm{perturb}}$.     $\triangleright \theta_{k,0}^z = \theta_k^z$

9:         estimate $\{\nabla_\theta \mathrm{L}_{\mathrm{SQ}}(\tilde{u}_{k,m}^s)\}_{m=1}^M$ by ZO in (14)

10:        update $\theta_{k,s+1}^z = \theta_{k,s}^z - \frac{\beta}{M} \sum_{m=1}^M \left( \nabla_\theta \mathrm{L}_{\mathrm{SQ}}(\tilde{u}_{k,m}^s) + \gamma \nabla_\theta \mathrm{L}_{\mathrm{SM}}(\theta_{k,s}^z, u_{k,m}) \right)$     $\triangleright \theta_{k+1}^z = \theta_{k,S_z}^z$

11:     **end for**

12:     calculate parameterization perturbation $q_{k,\mathrm{param}}^+ = q_{k,\mathrm{param}} + \nu q_{\mathrm{perturb}}, q_{k,\mathrm{param}}^- = q_{k,\mathrm{param}} - \nu q_{\mathrm{perturb}}$     $\triangleright \theta_{k,0}^z = \theta_k^z$

13:     calculate noise scheduler $q_k^+, q_k^-$ from $q_{k,\mathrm{param}}^+, q_{k,\mathrm{param}}^-$ by cosine or sigmoid parameterization

14:     sample $\{\tilde{u}_{k+1,m}^+, \tilde{u}_{k+1,m}^-\}_{m=1}^M$ from backward process (6) with $q_k^+, q_k^-$ and $\theta_{k+1}^z$.

15:     estimate $\nabla_{q_{\mathrm{param}}} \mathrm{L}_{\mathrm{SQ}}(\tilde{u}_{k+1,m}) = \frac{q_{k,\mathrm{perturb}}}{2\nu}(\mathrm{L}_{\mathrm{SQ}}(\tilde{u}_{k+1,m}^+) - \mathrm{L}_{\mathrm{SQ}}(\tilde{u}_{k+1,m}^-))$ by ZO

16:     calculate $\{\nabla_{q_{\mathrm{param}}} \mathrm{L}_{\mathrm{SM}}(\theta_{k+1}^z, u_{k,m}), \nabla_{q_{\mathrm{param}}} \mathrm{L}_{\mathrm{SM}}(\theta_{k+1}^y, u_{k,m})\}_{m=1}^M$ by auto-differentiation

17:     update $q_{k+1,\mathrm{param}} = q_{k,\mathrm{param}} - \frac{\eta_k}{M} \sum_{m=1}^M \left( \nabla_{q_{\mathrm{param}}} \mathrm{L}_{\mathrm{SQ}}(\tilde{u}_{k+1,m}) + \gamma(\nabla_{q_{\mathrm{param}}} \mathrm{L}_{\mathrm{SM}}(\theta_{k+1}^z, u_{k,m}) - \nabla_{q_{\mathrm{param}}} \mathrm{L}_{\mathrm{SM}}(\theta_{k+1}^y, u_{k,m})) \right)$

18:     update $q_{k+1,\mathrm{param}} = \mathrm{Proj}_\mathcal{Q}(q_{k+1,\mathrm{param}})$

19: **end for**

20: calculate noise scheduler $q_K$ from $q_{K,\mathrm{param}}$ by cosine or sigmoid parameterization

21: sample $\{\tilde{u}_{K,m}\}_{m=1}^M$ from the backward process (5) with $q_K$ and $\theta_K^z$.

22: **Output:** $(q_K, \{u_{K,m}^z\}_{m=1}^M)$.

---

original data distribution. The acquisition function used was the expected improvement (EI), defined as:

$$-EI(\lambda) = -\mathbb{E}[f(\lambda) - f(\lambda_t^+)]$$

where $f(\lambda_t^+)$ represents the best observed value at iteration $t$.

**Weighted sum.** For the weighted sum method, we jointly optimized the reward function and the CLIP score during the fine-tuning of the diffusion model. The optimization was performed by taking the weighted sum of the reward value and the CLIP score, with the weight for the CLIP score set to $0.5$. The $\lambda$ values used for the weighted sum method are selected by grid search with the search grid: $\lambda \in \{0.01, 0.1, 1.0, 10.0, 100\}$.

### F.2. Application 2: bilevel noise scheduling learning

This cosine parametrization in (16) covers both the cosine noise scheduler in (Nichol & Dhariwal, 2021) when $q_s = 0, q_e = q_\tau = 1$ and (Chen, 2023) when $q_\epsilon = 0$. Sigmoid parameterization is defined similarly by

$$l(t) = \mathrm{sigmoid}\left[ \frac{T - t(q_e - q_s) - q_s}{\tau T} + q_\epsilon \right] \tag{34}$$

which covers (Chen, 2023) when $q_\epsilon = 0$. Since $q(t)$ should be nondecreasing, we assign $q(t) = 1 - l(t)/l(t-1)$. With the use of parameterization, ZO perturbation will be added on $q_s, q_e, q_\epsilon, q_\tau$ instead of directly on $q(t)$.

For hyperparameter optimization for noise scheduler, we compare our method against greedy grid search, random search, Bayesian search and the default DDIM. Default parameter choices of DDIM with cosine noise scheduler in (Nichol & Dhariwal, 2021) are $q_s = 0, q_e = 1, q_\tau = 1, q_\epsilon = 0.008$. Default parameter choices of DDIM with sigmoid noise scheduler are $q_s = -3, q_e = 3, q_\tau = 0.1, q_\epsilon = -0.5$ according to (Vidhya, 2024).

**(Greedy) grid search.** According to the sensitivity analysis shown in Figure 6, the most sensitive parameter is $q_\tau$, while the last three almost equally important. Therefore, in greedy grid search, we tuned the parameters in the order $q_\tau, q_\epsilon, q_s, q_e$. We adopt the following search grid for cosine parametrization and best parameter given by greedy grid search is highlighted:

$$q_s \in \{0, \textbf{0.1}, 0.2, 0.3, 0.4\}$$
$$q_e \in \{\textbf{1}, 0.9, 0.8\}$$
$$q_\tau \in \{1, 2, \textbf{3}, 4\}$$
$$q_\epsilon \in \{0.005, 0.008, 0.01, \textbf{0.02}, 0.03, 0.04\}$$

For sigmoid parametrization, we use the following search grid and the best parameter is highlighted in black:

$$q_s \in \{-6, -5, \textbf{-4}, -3, -2, -1, 0\}$$
$$q_e \in \{2, \textbf{3}, 4\}$$
$$q_\tau \in \{0.1, 0.2, \textbf{0.3}, 0.4, 0.5, 1, 10\}$$
$$q_\epsilon \in \{-2, -1, \textbf{-0.5}, 0, 1\}$$

**Random search.** We sample 16 random combinations of $q_s, q_e, q_\tau, q_\epsilon$ from $q_s \in [0, 0.4], q_e \in [0.8, 1], q_\tau \in [1, 4], q_\epsilon \in [0.005, 0.04]$ for cosine parameterization and $q_s \in [-6, 0], q_e \in [2, 4], q_\tau \in [0.1, 1], q_\epsilon \in [-2, 1]$ for sigmoid parameterization. We report the best-performing results given by the random combination.

**Bayesian search.** We use the same range as the random search for Bayesian search and employ the same implementation to the first application.

**Bilevel algorithm.** We employ bilevel algorithm in Algorithm 6 and set the initialization of the noise scheduler parameter $q_s, q_e, q_\tau, q_\epsilon$ as the default values in DDIM (Nichol & Dhariwal, 2021; Vidhya, 2024). We use a batch size of 128 and choose the number of inner loop $S_z$ for $\theta^z$ updates as 1. Empirically, we found that, at the beginning of the training process (i.e. when $k = 0$), the number of inner loop $S_y^0$ for updating $\theta^y$ should be larger to obtain a relatively reasonable U-Net, but later on, we do not need large inner loop, i.e. we set $S_y^k = 10$ for $k \geq 1$. We formalize this stage as initial epoch, where we traverse every batch and set $S_y^0 = 20$. We choose the ZO perturbation amount as $\nu = 0.01$. Moreover, Algorithm 6 leverages the warm-start strategy.

**Warm-start strategy.** To further accelerate convergence, we avoid fully optimizing the penalty and lower-level problem with respect to $\theta$ for every $q$. Instead, we employ a warm-start strategy, initializing $\theta$ using its value from the previous epoch (Arbel & Mairal, 2022; Vicol et al., 2022; Sambharya et al., 2024). Empirically, this approach effectively reduces the inner loop for optimizing $\theta$ in the lower-level and penalty problems to 10 and 1, respectively. Moreover, only $3 - 4$ outer epochs are needed for optimizing $q$. Compared to the 100 epochs required for single-level diffusion model training, this significantly enhances the computational efficiency of our method, as shown in Table 3. With only $2.5\times$ the training time of a single-level diffusion model, the bilevel method achieves a 30% improvement over the default model while using just 15% of the time for Bayesian search.

**Exponential moving average (EMA).** We also incorporate EMA, which is an indispensable strategy in all high-quality image generation methods to stabilize training (Nichol & Dhariwal, 2021; Song et al., 2021b; Ho et al., 2022; Karras et al., 2022). EMA maintains a running average of model parameters over time, where recent updates are weighted more heavily than older ones so that it smooths out fluctuations in the training process.

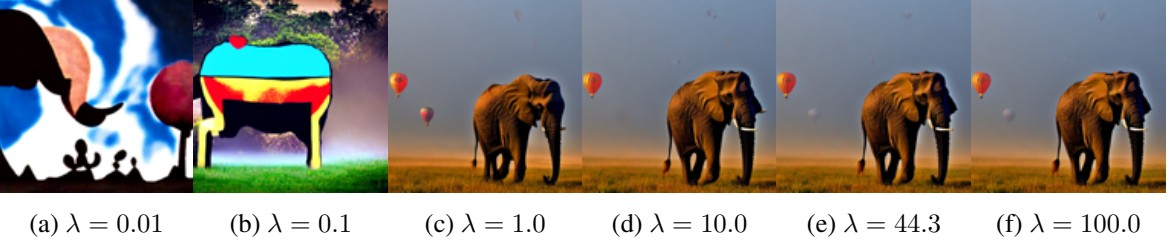

(a) $\lambda = 0.01$     (b) $\lambda = 0.1$     (c) $\lambda = 1.0$     (d) $\lambda = 10.0$     (e) $\lambda = 44.3$     (f) $\lambda = 100.0$

*Figure 8.* Balancing the realism and aesthetic in the image generation by controlling the entropy regularization strength parameter $\lambda$. Prompt: "An African elephant on a foggy morning, with hot air balloons landing in the background."

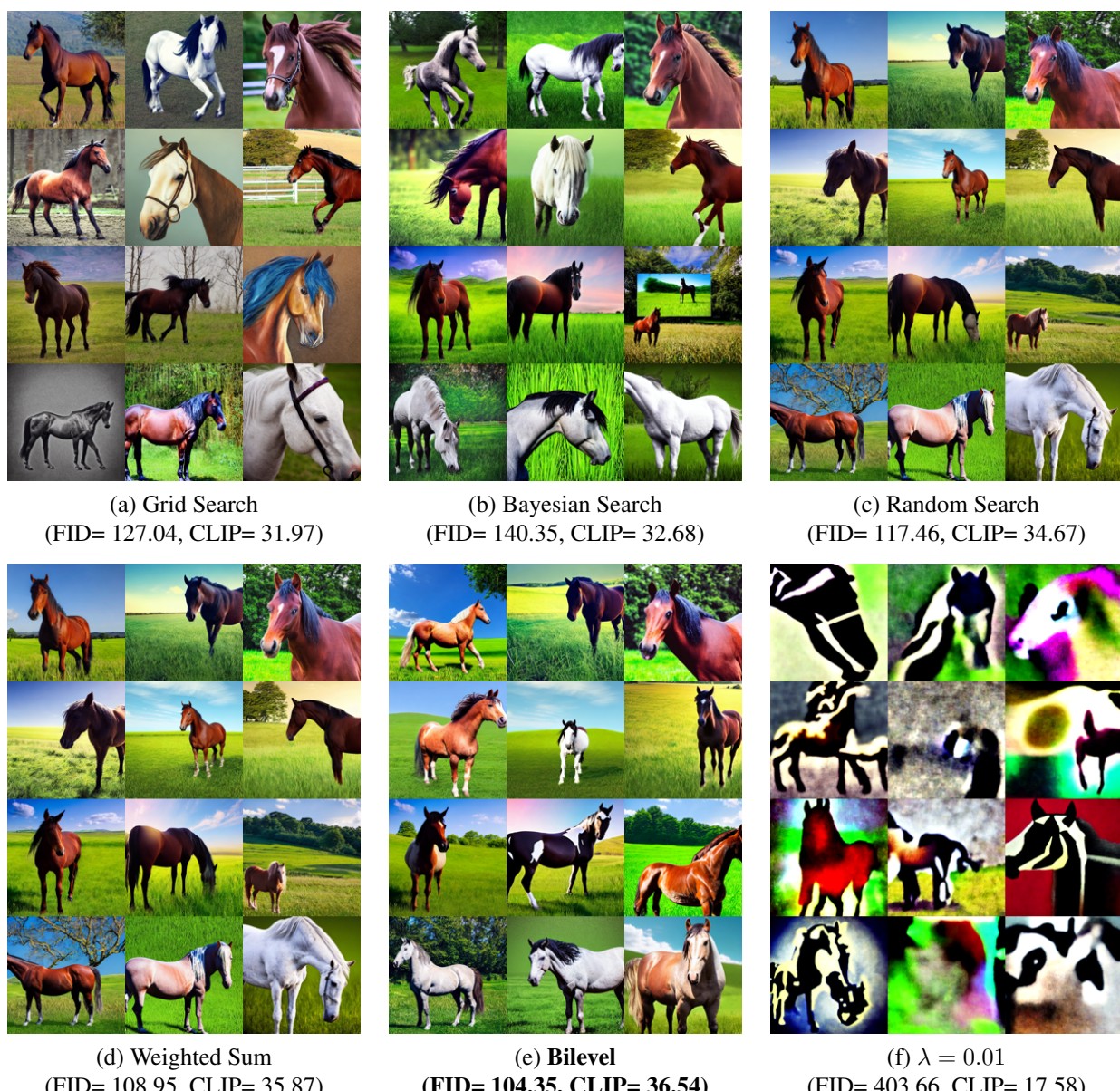

(a) Grid Search
(FID= 127.04, CLIP= 31.97)

(b) Bayesian Search
(FID= 140.35, CLIP= 32.68)

(c) Random Search
(FID= 117.46, CLIP= 34.67)

(d) Weighted Sum
(FID= 108.95, CLIP= 35.87)

(e) **Bilevel**
**(FID= 104.35, CLIP= 36.54)**

(f) $\lambda = 0.01$
(FID= 403.66, CLIP= 17.58)

*Figure 9.* Visualization of the final generated images (step-7) by different methods. Prompt: "A realistic photo of a horse standing on lush green grass in a countryside meadow on a sunny day, with clear blue sky in the background." **(a) Grid Search:** The generated images do not fully adhere to the prompt, as the clear blue sky is often missing. Some images appear more abstract. **(b) Bayesian Search:** Most images lack a blue sky in the background, and some horses are deformed. **(c) Random Search:** In certain images, the mane is not well-defined. **(d) Weighted Sum:** Some images exhibit imperfections in the mane and facial features of the horses. **(e) Bilevel:** Generates visually striking, highly realistic images that closely align with the given prompt.

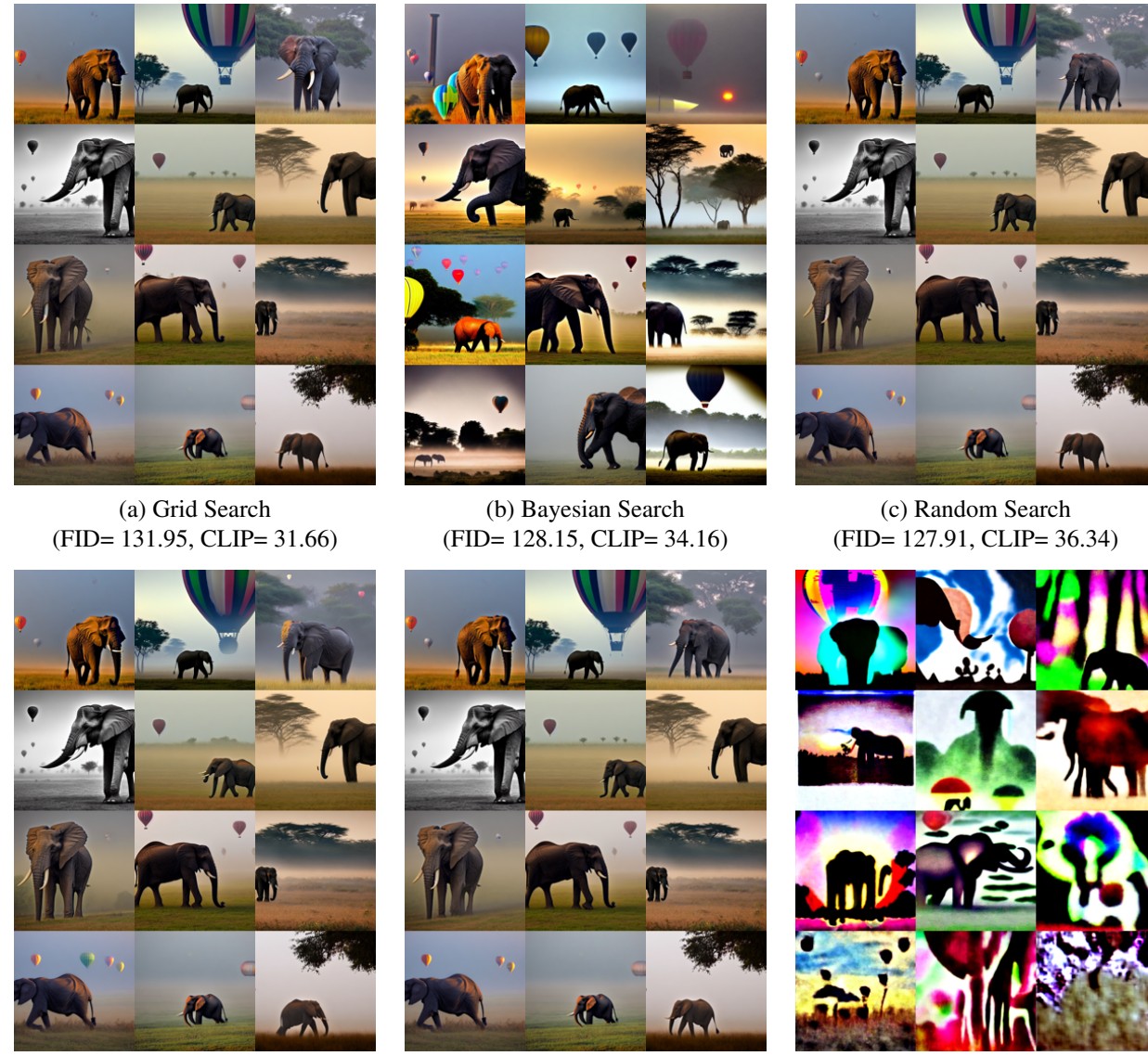

(a) Grid Search
(FID= 131.95, CLIP= 31.66)

(b) Bayesian Search
(FID= 128.15, CLIP= 34.16)

(c) Random Search
(FID= 127.91, CLIP= 36.34)

(d) Weighted Sum
(FID= 117.95, CLIP= 37.85)

(e) **Bilevel**
**(FID= 105.72, CLIP= 39.05)**

(f) $\lambda = 0.01$
(FID= 440.40, CLIP= 18.78)

*Figure 10.* Visualization of the final generated images by different methods. Prompt: "An African elephant on a foggy morning, with hot air balloons landing in the background." **(a) Grid Search:** Some images exhibit deformed elephant figures, and the hot air balloons are missing. **(b) Bayesian Search:** The images appear more abstract, with deformed elephants and trees. Elephant is even missing in one figure, while another depicts elephants on top of a tree. **(c) Random Search:** The elephant and hot air balloons are unclear in some images. **(d) Weighted Sum:** Struggles to generate a recognizable elephant figure, producing a deformed trunk instead. **(e) Bilevel:** Generates relatively high-quality images with no visible deformations.

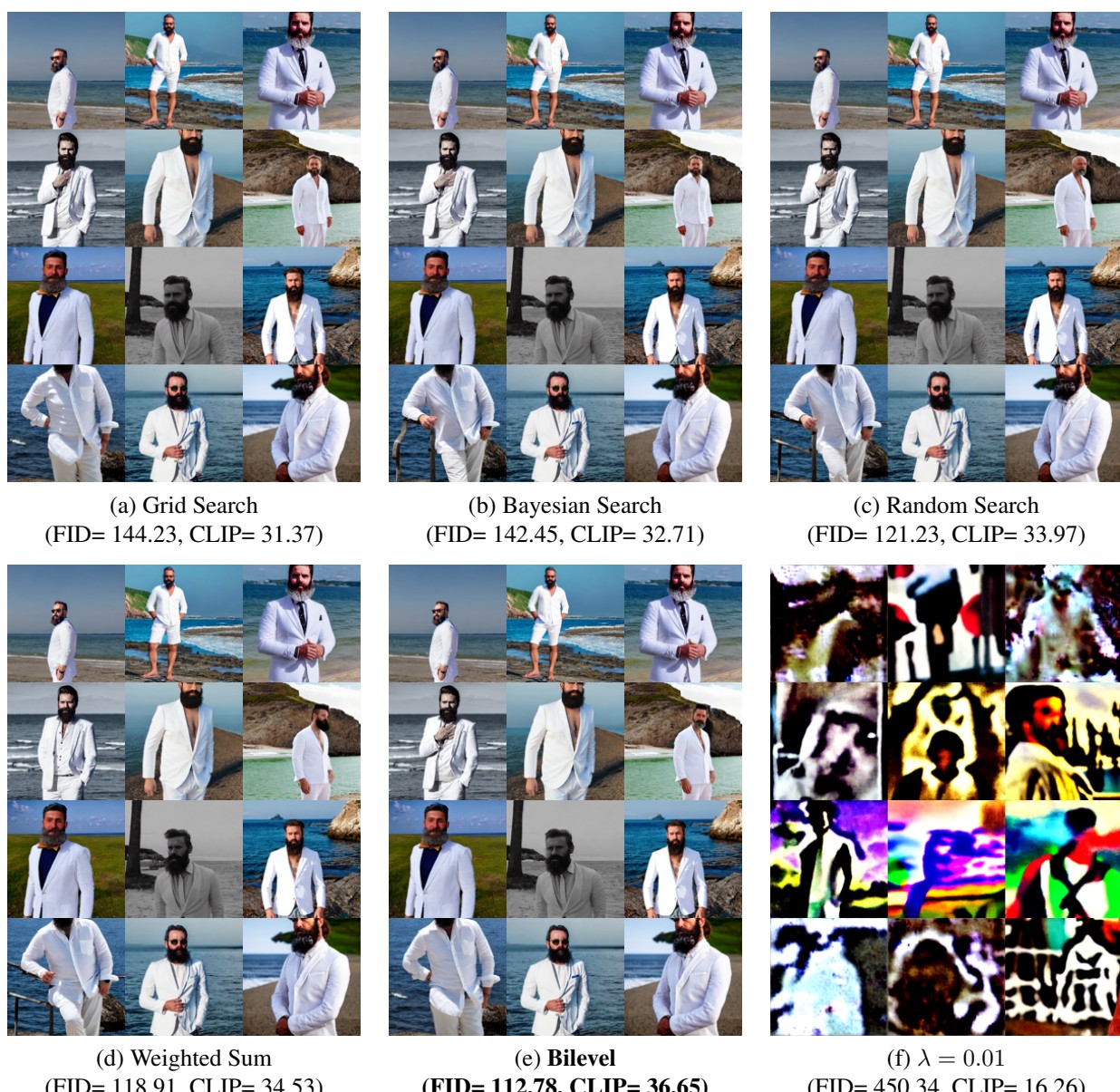

(a) Grid Search
(FID= 144.23, CLIP= 31.37)

(b) Bayesian Search
(FID= 142.45, CLIP= 32.71)

(c) Random Search
(FID= 121.23, CLIP= 33.97)

(d) Weighted Sum
(FID= 118.91, CLIP= 34.53)

(e) **Bilevel**
**(FID= 112.78, CLIP= 36.65)**

(f) $\lambda = 0.01$
(FID= 450.34, CLIP= 16.26)

*Figure 11.* Visualization of the final generated images by different methods. Prompt: "A gentleman wearing white clothes and a beard, posing in a seaside setting." **(a) Grid Search:** Struggles to generate human faces; images appear blurry. **(b) Bayesian Search:** Some faces are blurry, and hands are deformed. **(c) Random Search:** Some face images appear blurry. **(d) Weighted Sum:** Produces comparatively good-quality images. **(e) Bilevel:** Generates relatively high-quality images with no visible deformations.

