# OpenReview forum: "A First-order Generative Bilevel Optimization Framework for Diffusion Models"
_ICML.cc/2025/Conference — ICML 2025 poster_

### Official Review · Reviewer_ExjZ · 2025-03-10

**Overall Recommendation:** 4

**Summary:**

The authors have proposed a bilevel optimization framework tailored for diffusion models, specifically addressing two scenarios:-
Fine Tuning a pre-trained diffusion models (via Inference-only Solver):

To fine-tune pre-trained diffusion models, to maximize the task-specific rewards, while preserving the aesthetic realism, the authors propose a bilevel optimization problem as:
- Upper-level problem: Select optimal hyperparameters such as entropy regularization strength λ, that balances reward maximization and realism.
- Lower-level problem: Adjusts the generated data distribution to maximize a reward function with entropy regularization, ensuring closeness to the pre-trained distribution.
This method avoids expensive backpropagation through diffusion steps by using guided sampling and closed-form gradient estimation.

Noise Schedule Optimization (Training from Scratch):
The authors optimize noise schedules used during training diffusion models, and the bilevel problem is defined as:
- Upper-level problem: Optimizes parameters controlling the noise schedule to minimize metrics like FID score of the generated images.
- Lower-level problem: Learn parameters of a score function that approximates gradients of log-likelihoods of noisy data distributions.
To efficiently solve this nested structure without differentiating through multiple sampling steps (which would be computationally intensive), the authors use:
- Reparameterization of noise schedules (using cosine or sigmoid functions with only four parameters).
Zeroth-order gradient estimation, allowing gradient approximation without explicit backpropagation through sampling trajectories


The authors convert nested bilevel optimization problems into a single-level penalty problems solvable via first-order methods, providing theoretical guarantees under strong convexity assumptions

**Claims And Evidence:**

Proposition 1 derives a closed-form gradient estimator for entropy regularization strength (λ) using pre-trained data samples, avoiding backpropagation through diffusion steps. Theorem 1 establishes convergence guarantees under strong convexity assumptions (Assumption 1) Reformulates bilevel problems into single-level penalty objectives (Eq. 2), enabling gradient updates via inference-only guided sampling (Alg. 5)

**Essential References Not Discussed:**

NA

**Experimental Designs Or Analyses:**

Experimental Setup:

For pre-trained models, reward finetuning, the authors use StableDiffusion V1.5 model as their pre-trained model and employ a ResNet-18 architecture (trained on the ImageNet dataset) as the synthetic (lower-level) reward model. The bilevel method achieved an 11.76% improvement in the FID score and an 8.32% improvement in the CLIP score over the best-performing weighted sum method.

For Noise Schedule Optimization, the authors train a U-Net model on MNIST from scratch, and use Cosine/sigmoid schedules with 4 parameters. The authors claim their bilevel method achieved 30% lower FID than default DDIM with only 2.5× training time.

**Methods And Evaluation Criteria:**

Methods: The authors have proposed a Bilevel Optimization for guidance of diffusion models to maximize the reward function, at the same time, to maintain the realism of the generated images. They have proposed this method for two scenarios: (i) inference based fine tuning of pretrained models, (ii) noise schedule optimization when training from scratch.

Evaluation Criteria: The authors use FID score, CLIP score, IS score and time (is seconds) to discuss the effectiveness of their proposed method.

**Other Comments Or Suggestions:**

The images provided by the authors are a bit difficult to understand in terms of their aesthetics. If the authors can put 1 image per prompt rather than a collage, will help to differentiate their methods effectiveness.

**Other Strengths And Weaknesses:**

**Strengths**

Theoretical Innovation:
- Closed-form gradient estimation: Proposition 1 enables direct computation of gradients for entropy regularization strength (λ) using pre-trained samples, eliminating backpropagation through diffusion steps.
- Convergence guarantees: Theorem 1 establishes convergence under strong convexity assumptions, matching standard bilevel optimization rates.

Algorithmic Design:
- Inference-only fine-tuning: Avoids backpropagation through sampling trajectories by using guided sampling (Algorithm 5), reducing memory costs.
- Parameterized noise schedules: Reduces noise schedule optimization to 4 parameters (e.g., cosine/sigmoid functions) instead of tuning per-step values.
- Zeroth-order (ZO) gradients: Enables gradient estimation for noise schedules without differentiating through sampling steps (Equation 14).

**Weaknesses**

- Baseline Comparisons: The authors should compare their bi-level optimization approach with other reward model based approaches (mentioned above), to see how much their method performs.
- Reproducibility: The authors should release the code, and their experimental setup, to support their work.

**Questions For Authors:**

- Is there a relaxation for strongly convex setup, which might be applicable here. If a better proof can be provided for the same, would strengthen their method.
- Memory cost comparison can be further explained, as it has been mentioned in few of the places.
- Can the authors comment further on other related works (as mentioned above), in terms of aesthetics, computation, methods, and comparison metrics?

**Relation To Broader Scientific Literature:**

The authors claim, this is the first work related bilevel optimization to diffusion models. Their work fits into the broader space of AI Alignment, and Reward based optimization of generative models. Their work is foundational towards understanding alignment from a theoretical perspective.

The authors discuss reward alignment from bilevel optimization perspective. There are few recent papers on this topic, such as:-

1. Implicit Diffusion: Efficient optimization through stochastic sampling
2. SPARKLE: A Unified Single-Loop Primal-Dual Framework for Decentralized Bilevel Optimization
3. Bi-level Guided Diffusion Models for Zero-Shot Medical Imaging Inverse Problems

Apart from that, in a broader sense, since the method is towards inference time alignment as well, the authors can comment on other test time alignment and reward based methods such as:-

1. Aligning Text-to-Image Diffusion Models with Reward Backpropagation
2. Aligning Diffusion Models with Noise-Conditioned Perception
3. Diffusion Model Alignment Using Direct Preference Optimization

**Theoretical Claims:**

Theorem 1 establishes convergence guarantees for the proposed bilevel optimization framework under the following conditions:
- Strong convexity: The lower-level objective is strongly convex.
- Smoothness: f(x,y) and g(x,y) are jointly smooth over (x,y) with constants l_{f,1} and l_{g,1}.
- Lipschitz continuity: f(x,⋅) is l_{f,0} 0-Lipschitz, and g(x,y) has l_{g,2} Lipschitz Hessian.

Under Assumption 1, the bilevel algorithm ensures a strict descent in the upper-level objective F(x). This theorem bridges theory and practice, ensuring that the framework’s hyperparameter updates (e.g., entropy strength λ or noise schedule parameters) provably guide diffusion models toward better performance under realistic Assumptions1.

---

> ### Author Rebuttal · Authors · 2025-03-31
>
> We thank the reviewer for appreciating the theoretical innovation and our algorithm design. Our response to your comments follows.
>
> **Q1. Baseline comparison.** Thank you for your question. We have added numerical comparisons using different reward functions during the rebuttal period; please see Table R1 and Figure R1 on anonymous link:  https://anonymous.4open.science/r/bilevel-diffusion-11A1/bilevel_diffusion_rebuttal.pdf. We test the performance of our method for another widely used lower-level reward function, HPSv2. Bilevel approach also outperform other baselines in terms of image quality in comparable time complexity, which showcase the robustness of our approach with respect to different reward functions.  Due to time constraints, we could not include all the baselines you mentioned, as they all correspond to the lower-level reward fine-tuning task and require additional HPO on top of that. We believe these methods, though differing in their lower-level fine-tuning strategies, could similarly benefit from an upper-level HPO. We will discuss these points further in our revised manuscript.
>
> **Q2. Reproducibility.** Thank you for your suggestion. We will release it with the final version.
>
> **Q3. Presentation of images.** Thank you for your suggestion. We will change accordingly.
>
> **Q4. Relaxation of strongly convexity assumption.** Thank you for your question. Yes, it is possible to relax strongly convexity assumption to so-called Polyak-Lojasiwcz condition following (Kwon et. al 2024, Shen et. al 2024). This condition covers nonconvex objectives and are satisfied by loss of overparameterized neural network [R5].
>
> > [R5] Loss landscapes and optimization in over-parameterized non-linear systems and neural networks. C. Liu, et al. 2022.
>
> **Q5. Memory cost comparison.** For the first application of reward fine-tuning, since Algorithm 2 seprates the hyperparameter optimization stage and sampling stages, it does not introduce additional memory cost. For the second application of noise scheduling, we add a Table R2 in attached PDF to compare the memory usage of bilevel HPO and other approaches. By utilizing ZO and noise scheduler parameterization, the memory overhead of bilevel method is not severe.
>
> **Q6. Comparison with related works.** Thank you for providing these literatures. Most of related works are related to the lower-level task for the first application on the fine-tuning diffusion model stage, including [R6-R11]. They belong to related works in "fine-tuning diffusion models" section and we adopt the guidance based fine-tuning method for the lower-level task. All of the methods in [R6-R11] can be further benefits from tuning the KL regularization using our framework. We will add a discussion in the paper.
>
>
> > [R6] Implicit Diffusion: Efficient optimization through stochastic sampling
> > [R7] SPARKLE: A Unified Single-Loop Primal-Dual Framework for Decentralized Bilevel Optimization
> > [R8] Bi-level Guided Diffusion Models for Zero-Shot Medical Imaging Inverse Problems
> > [R9] Aligning Text-to-Image Diffusion Models with Reward Backpropagation
> > [R10] Aligning Diffusion Models with Noise-Conditioned Perception
> > [R11] Diffusion Model Alignment Using Direct Preference Optimization

---

> > ### Comment · Reviewer_ExjZ · 2025-04-04
> >
> > Thanks for the clarification, this addresses my remaining concerns.

---

> > > ### Author Response · Authors · 2025-04-06
> > >
> > > Thank you for taking the time to review our work and response, and for providing constructive suggestions.

---

### Official Review · Reviewer_FnLj · 2025-03-12

**Overall Recommendation:** 4

**Summary:**

This paper explores the application of bilevel optimization in diffusion models, focusing on two key applications. The first optimizes the trade-off parameter that balances the reward and proximity to the pre-trained distribution during fine-tuning. The second optimizes the noise schedule in diffusion models. To enhance scalability, the paper proposes a first-order optimization framework specifically designed for diffusion models. Experimental results demonstrate that the proposed method significantly reduces hyperparameter optimization time compared to grid search, random search, and Bayesian optimization.

**Claims And Evidence:**

The proposed method is theoretically well-founded and demonstrates strong empirical performance.

**Essential References Not Discussed:**

No

**Experimental Designs Or Analyses:**

Yes.

**Methods And Evaluation Criteria:**

The proposed method adapt the existing fully-first order method to diffusion model which makes sense.

**Other Comments Or Suggestions:**

In eq 1, should the optimization variable in upper level be only x?

**Other Strengths And Weaknesses:**

The proposed method is theoretically grounded and specifically designed for diffusion models, demonstrating strong empirical performance.

**Questions For Authors:**

Could the method be extended to the setting whether the upper-level objective is not differentiable?

**Relation To Broader Scientific Literature:**

The proposed method builds on the fully first-order approach from Kwon et al. (2023) and introduces improvements to enhance its scalability for diffusion models

**Theoretical Claims:**

I check the proofs of proposition 1 and theorem 1.

---

> ### Author Rebuttal · Authors · 2025-03-31
>
> We thank the reviewer for appreciating the theoretical guarantees for our algorithm and its empirical performance. Our response to your comments follows.
>
> **Q1. Optimization merely on $x$ in equation 1?**
>
> Thank you for your question. In equation 1, we state the general bilevel HPO formulation where the lower-level objective is not necessarily strongly convex so that $\mathcal{S}(x)$ may contain multiple solutions. Therefore, we also need to optimize on $y$ to select one solution. To eliminate any confusion, we will revise our formulation throughout to assume a strongly convex lower-level objective so that optimizing $x$ merely is enough.
>
> **Q2. Applicability of method on non-differentiable objective.**
>
> Thank you for your question. Our approach in the second application can extend to non-differentiable upper-level metrics, as ZO estimation is compatible with non-differentiable objectives. For the first application, if the reward function is non-differentiable, we would similarly employ a ZO estimator for its gradient; see [R4].
>
> > [R4] An Algorithm with Optimal Dimension-Dependence for Zero-Order Nonsmooth Nonconvex Stochastic Optimization. G. Kornowski, O. Shamir. JMLR, 2024.

---

### Official Review · Reviewer_XKpE · 2025-03-13

**Overall Recommendation:** 3

**Summary:**

This paper introduces a practical, first-order bilevel framework for diffusion models, outperforming standard methods in fine-tuning and training scenarios. The proposed method eliminates the high dimensionality and sampling costs in traditional methods.

**Claims And Evidence:**

The claims are supported by experimental results and theoretical convergence guarantees.

**Essential References Not Discussed:**

References are good to me.

**Experimental Designs Or Analyses:**

Since the main objective of this paper is to address practical challenges in diffusion models, the numerical experiments presented are insufficiently representative.

Specifically, Section 6.1 optimizes only one hyperparameter, resulting in limited improvement over naive search methods.

Additionally, the experiments in Section 6.2 rely solely on the MNIST dataset, which might be simple for large-scale cases for diffusion model applications and thus does not adequately reflect the performance of modern diffusion frameworks. More large-scale datasets could be added to make the results more convincing.

**Methods And Evaluation Criteria:**

Yes, the proposed method and evaluation criteria make sense.

**Other Comments Or Suggestions:**

Line 128 left, $S_\gamma^*(x)$ is not defined.

Line 201 right, $y^*$ and $z^*$ are mismatched.

Line 437 right, the quotation marks need some adjustments.

**Other Strengths And Weaknesses:**

The application of bilevel optimization to tuning hyperparameters in diffusion models is novel and interesting. From the bilevel optimization side, the novelty is not very significant. The paper could be strengthened if more comprehensive and large-scale experiments can be done.

**Questions For Authors:**

Although I appreciate the novel application of gradient-based hyperparameter tuning for diffusion models. I still want to understand from the optimization perspective, what new challenges the diffusion models introduce to the algorithmic development in bilevel optimization? In other words, what is the main technical or algorithmic novelty on the bilevel optimization side?

**Relation To Broader Scientific Literature:**

The main contribution of this paper is framing diffusion hyperparameter tuning as a bilevel optimization problem.

On the positive side, this formulation appears novel in the diffusion context. However, the motivation is not that exciting because hyperparameter tuning is already a standard application of bilevel optimization. A more promising direction may involve using a bilevel optimization approach to address fundamental diffusion challenges, such as improving sampling efficiency.

**Theoretical Claims:**

The main proof is provided in D.3 and it is correct.

---

> ### Author Rebuttal · Authors · 2025-03-31
>
> We thank the reviewer for appreciating our novelty. We hope our response to your comments below can resolve your minor concerns.
>
> **Q1. Insufficient numerical experiments.**
>
> - **One hyperparameter in the fine-tuning diffusion model experiment.** Although KL regularization is just one hyperparameter, careful tuning it is essential. The appropriate KL strength $\lambda$ prevents reward over-optimization on downstream tasks by keeping the model close to the pre-trained distribution, while still allowing for the necessary variability to improve the reward (see Uehara et al., 2024; Fan et al., 2024) and Figure 2. We also consider multiple hyperparameter tuning on the second experiment for noise scheduling in training diffusion model.
> - **Noise scheduling tasks only on MNIST dataset.** The noise scheduling problem we consider arises during the (pre-)training stage of diffusion models, which is particularly expensive compared to fine-tuning. Notably, the cost of training a diffusion model on MNIST is already comparable to the cost of fine-tuning on ImageNet (see Tables 1 and 2). Moreover, training requires tuning multiple hyperparameters in noise scheduler parameterization, further increasing the computational burden (see grid search method in Table 2). Our goal for the second application is to propose a new automatic HPO framework for training diffusion models via bilevel optimization, so we demonstrate our method using the MNIST dataset in this version. More datasets will be investigated in future work.
> - **More comprehensive experiments.** During the rebuttal period, we add more comparisons on HPO for the fine-tuning diffusion model task. We test the performance of our method for another widely used lower-level reward function, HPSv2. See Table R1 and Figure R1 on anonymous link:  https://anonymous.4open.science/r/bilevel-diffusion-11A1/bilevel_diffusion_rebuttal.pdf. Bilevel approaches also outperform other baselines in terms of image quality in comparable time complexity, which showcase the robustness of our approach with respect to different reward functions.
>
> **Q2. New challenges and novelty from the optimization perspective.**
>
> Thank you for your question. As we highlighted at the end of Section 1, there are *sufficient novelty* in the context of bilevel optimization as well. The *first challenge* is that we cannot directly optimize the distribution itself but can only work with samples. In HPO for fine-tuning a diffusion model, we use guided backward sampling to generate samples approximately from the desired distribution. Thanks to (Guo et al, 2024), we bridge the gap between guided sample-generation guarantees and optimization guarantees for the underlying probability distribution in this settings. In HPO for training a diffusion model, we parameterize the noise scheduler and noise distribution via  cosine/sigmoid functions and a score network, respectively, and optimize their parameters instead. The *second challenge* is the numerical feasibility and computational overhead. For Application 1, we derived a backward-process-free approach to estimate the upper-level gradient (Proposition 1), reducing complexity. For Application 2, we employ ZO estimation to avoid both computational and memory costly backpropagation. Finally, we validate the assumption of strong convexity in probability space for diffusion models which are discussed in "implications for generative bilevel applications" in Section 5.
>
> **Q3. Hyperparameter tuning is a standard application of bilevel optimization.**
>
> Thank you for your question. Note that although prior work has explored HPO via bilevel optimization, most existing methods rely on implicit gradient or unrolling differentiation approaches that entail costly second-order computations. In contrast, we leveraged a fully first order bilevel method, *which is new in HPO*. Moreover, the techniques in existing works do not readily extend to the infinite-dimensional probability space of diffusion models. The new challenges we highlight in Q2 are fundamental when applying bilevel HPO to diffusion models.
>
> **Q4. Applicability of noise scheduling tuning in faster sampling with a diffusion model.**
>
> We are happy to see our work may stimulate promising direction! This is indeed an interesting future direction for us. We note the emerging line of work on noise optimization for faster fine-tuning (e.g., Tang et al., 2024), but the stages they target differ from ours. Those methods focus on reward fine-tuning using a pre-trained model so that it is essentially a single-level reward maximization task, similar to non-HPO version of our first application on reward fine-tuning, whereas we target automatic HPO for the fundamental diffusion model training stage. Moreover, as our method explored the better choice of noise scheduler at each iteration, our method can potentially reduce the overall sample complexity needed to achieve the target image quality in diffusion model training.

---

### Official Review · Reviewer_8QSP · 2025-03-14

**Overall Recommendation:** 3

**Summary:**

The paper explores the problem of bilevel optimization with diffusion models - a hierarchical framework consisting of a higher and lower level objectives which are jointly optimized. The authors frame the following two problems as bilevel optimization:
1. KL regularized reward maximization as the lower level objective for diffusion model finetuning, and the higher level objective optimizing the KL weight $\lambda$ using CLIP as the reward.
2. Tuning the noising schedule of a diffusion model in the upper level and score matching loss in the lower level.
For the reward guided finetuning task, the authors use a color/vibrancy reward for the lower level objective and CLIP score for the higher level objective, and demonstrate improvement over other search methods such as grid search and Bayes opt. For the second task, the authors finetune the noise schedule of an MNIST generative model using DDIM inference.

**Claims And Evidence:**

The paper claims to propose an efficient framework to perform bilevel optimization with diffusion models. I think the claims for the most part are justified by the experiments, but I have not gone through the implementation details thoroughly enough to know if the methods being compared against are fairly tuned.

**Essential References Not Discussed:**

No essential references are missing to my knowledge.

**Experimental Designs Or Analyses:**

I did not notice any issues with the experimental design.

**Methods And Evaluation Criteria:**

I am not fully convinced that the two tasks the paper explores are directly useful themselves. Usually in previous work, entropy regularized reward guided sampling of diffusion models (the lower level problem) is investigated in isolation. I can see the utility of tuning the KL weight, but the complexity added with bilevel optimization doesn't seem useful in practice. However, it is a reasonable experiment to demonstrate the method in the context of this paper. The noise scheduler optimization with score matching is a lower level loss is also a somewhat confusing experiment, and only demonstrating qualtitative results on MNIST is a weak result.

**Other Comments Or Suggestions:**

NA

**Other Strengths And Weaknesses:**

### Weaknesses
1. I found the paper quite confusing to read throughout. After reading the paper, I still do not fully understand the motivation.

**Questions For Authors:**

1. Could the authors clarify the two issues I brought up regarding theoretical claims?
2. Is there any reason the noise scheduling task was only done for MNIST sampling? It would be more convincing if done on with a more difficult dataset.

**Relation To Broader Scientific Literature:**

I am not aware of previous works that are directly related to the proposed bilevel optimization framework. The noise scheduling task could be applicable to faster sampling with a diffusion model. The bilevel reward finetuning task is potentially more relevant in my opinion, and could we used alongside other strategies from the diffusion guidance [1] for solving the low level optimization task.

[1] Inference-Time Alignment in Diffusion Models with Reward-Guided Generation: Tutorial and Review, https://arxiv.org/abs/2501.09685

**Theoretical Claims:**

I did not thoroughly check the theoretical claims and math, however I did notice a couple of potential issues:
1. The gradient in Proposition 1, is not correctly estimated using the MC estimator in Appendix E.2. The expectation is inside the log, so taking MC average inside the log would provide a biased estimator. However, section 4.1 seems to imply we can tractably estimate this gradient.
2. In page 4, final paragraph (section 4.1) it is written that algorithm 5 sampled from the optimal entropy guided distribution. But this is also an intractable task, and I believe the sampling process would only provide approximate guidance.

---

> ### Author Rebuttal · Authors · 2025-03-31
>
> We thank the reviewer for appreciating our topic. We hope our response to your comments below can resolve your minor concerns.
>
> **Q1. Motivation of two hyperparameter optimization (HPO) problems.**
>
> In the first application, the primary concern is whether adding additional computational cost for HPO is worthwhile, whereas in the second application, the concern is on its rationale.
>
> - **First application - Reward fine-tuning**. Our research is complementary to the entropy-regularized reward-guided diffusion models since no matter how to choose the guidance terms, the tuning of KL regularization is inevitable. Conventionally, tuning $\lambda$ via cross-validation (e.g. grid search) also incurs computational overhead due to the lack of universal guidelines across datasets and prompts (Uehara et al., 2024; Fan et al., 2024).  In contrast, reward-guided diffusion model associated with our bilevel-based HPO approach yields better FID and CLIP scores while preserving time complexity (see Table 1 and the 3rd paragraph in Section 6.1). Furthermore, without bilevel, a common strategy to use CLIP for enhancing realistic is to include it in a weighted sum alongside the reward guidance, which requires the costly CLIP gradient. In contrast, our bilevel method, thanks to Prop. 1, is CLIP gradient-free, which cuts down overall time costs; see Table 1.
>
> - **Second application -  Noise scheduling**. The noise scheduler, the variance of the added noise, controls the generation quality in diffusion models. If the variance is too large, the forward process quickly becomes noise and fails to learn meaningful representation for backward generation; if too small, it never fully reaches Gaussian noise. Prior work has highlighted the need to tune this noise scheduler to balance this trade-off (Nichol & Dhariwal, 2021; Lin et al., 2024; Chen, 2023; R1). In our bilevel HPO framework, we first fix the noise scheduler $q(t)$ and train the diffusion model using score-matching function as in (Song et al., 2021a,b; Ho et al., 2020; Nichol et al., 2021), then measure backward image quality via FID on the upper-level. The proposed method not only outperforms and significantly accelerates runtime of conventional HPO, but also enhances DDIM with empirically chosen schedules in comparable time; see Table 2.
>
> > [R1] Simple diffusion: End-to-end diffusion for high resolution images. E Hoogeboom, et. al. ICML 2023.
>
> **Q2. Biased Montel Carlo (MC) estimation.** Yes, the vanilla MC estimator is biased, but Theorem 1 can accommodate $\epsilon_k$ error. Moreover, since the reward is always positive, we have $e^{r_2(\cdot)/\lambda}, e^{(r_1(\cdot) / \gamma+r_2(\cdot))/\lambda}\geq 1$. Then since $\log(a)$ is Lipschitz continuous when $a\geq 1$, the MC estimation error can be controlled by large sampling batch size (see (Ji et al. 2021; Arbel & Mairal, 2022; Ghadimi & Wang, 2018)) or momentum updates (see [R2,R3]). However, empirical evidence suggests that we do not need a very large batch size and gradient updates without momentum also work well. We will add a comment in the revision and leave the theoretical improvement based on momentum updates for future work.
>
> > [R2] Stochastic compositional gradient descent: algorithms for minimizing compositions of expected-value functions. M. Wang, et. al. Math. Programming, 2017.
>
> > [R3] Solving stochastic compositional optimization is nearly as easy as solving stochastic optimization. T. Chen, et. al. IEEE Trans. on Signal Processing, 2021.
>
>
> **Q3. Generating from optimal entropy guided distribution.**
>
> We only need generation from the $\epsilon$-optimal entropy guided distribution and the error in Algorithm 5 can be accommodated by $\epsilon_k$ in Theorem 1. Moreover, Algorithm 5 in (Guo et al, 2024) converges linearly under a concave reward function and a strongly convex regularization term, implying that this accuracy can be achieved in $\mathcal{O}\bigl(\log(\epsilon_k^{-1})\bigr)$ iterations of Algorithm 5, which is not a huge computational burden. We will add a comment in the revision.
>
>
> **Q4. Applicability of other guidance strategies.**
>
> Yes, other guidance terms can be applied as long as they ensure the backward process converges on the optimal entropy-guided distribution, since our entropy-weight tuning approach is guidance-agnostic. We specifically choose the guidance from (Guo et al., 2024) because of its finite-time optimization guarantees. A clarifying paragraph will be provided.
>
> **Q5. Applicability of the noise scheduling task in faster sampling with a diffusion model.**
>
> Due to the space limit, please see the response to **Q4** for Reviewer XKpE.
>
> **Q6. Noise scheduling task on MNIST dataset.**
>
> Due to the space limit, please see the response to **Q1** for Reviewer XKpE.
>
> **Q7. Writing of this paper.**
>
> We follow the bilevel HPO framework, so some diffusion model details are moved to the Appendix due to space limit. We will clarify and improve the writing in the revision.

---

> > ### Comment · Reviewer_8QSP · 2025-04-04
> >
> > I thank the authors for their response, and for clarifying some points. Fundamentally, I think the paper isn't improved by these answers for me, so I will keep my score

---

> > > ### Author Response · Authors · 2025-04-06
> > >
> > > Thank you for taking time to review our clarifications and providing constructive feedback. We will consider your insights carefully in the revision as we continue to improve our work.

---

### Decision · Program_Chairs · 2025-05-01

**Decision:**

Accept (poster)

**Comment:**

This paper proposes a first-order bilevel optimization framework tailored to diffusion models, with two main applications: (1) fine-tuning pretrained diffusion models via reward-guided sampling by optimizing a KL regularization weight, and (2) optimizing noise schedules during training from scratch to improve generation quality. To ensure practical tractability, they avoid backpropagation through the generative process and use guidance for instance in (1). Experiments are conducted on Stable Diffusion (reward tuning) and MNIST (noise schedule tuning), showing improved sample quality and efficiency over baseline HPO methods like grid search and Bayesian optimization.
Strengths:
* Novel Application of Bilevel Optimization: While bilevel optimization is a known tool in hyperparameter tuning, applying it directly and efficiently to diffusion models is relatively novel and non-trivial.
* Practical Efficiency: The authors convincingly argue that their approach reduces the computational cost of tuning reward weights and noise schedules in diffusion settings. Tables and comparisons with traditional methods support this.
Weaknesses:
* Limited Experimental Scope: A recurring concern among reviewers (especially XKpE and 8QSP) was that experiments, particularly for the noise scheduling task, were confined to MNIST.  The authors specify that noise-tuning/training on mnist is as costly as fine-tuning on imagenet.
* Writing Clarity and Motivation: Some reviewers found the motivation and technical exposition confusing or unclear (notably 8QSP). The complexity of bilevel optimization combined with diffusion model details makes parts of the paper hard to follow.
* Limited Differentiation on the Optimization Side: While the application to diffusion models is novel, reviewers noted that the optimization algorithm itself is an adaptation of existing first-order bilevel methods. More innovation on this front could strengthen the contribution. Also, the theoretical results (eg Theorem 1) appear to be standard in bilevel optimization.
All reviews were positive (3/5,3/5,4/5,4/5) and remained so after rebutal.
While the paper's core contribution lies more in its application of bilevel optimization than in fundamentally new algorithmic techniques and theoretical contributions, this application is meaningful, non-trivial, and supported by both theory and experiments. The authors have thoughtfully addressed reviewer feedback provided additional experiments.
With additional large-scale experiments and clearer exposition, the paper would be stronger, yet in its current state it could make a notable contribution to both the diffusion modeling and bilevel optimization communities. It is therefore recommended for acceptance.